# Mutations in CFAP43 and CFAP44 cause male infertility and flagellum defects in *Trypanosoma* and human

Charles Coutton *et al.*[#]

Spermatogenesis defects concern millions of men worldwide, yet the vast majority remains undiagnosed. Here we report men with primary infertility due to multiple morphological abnormalities of the sperm flagella with severe disorganization of the sperm axoneme, a microtubule-based structure highly conserved throughout evolution. Whole-exome sequencing was performed on 78 patients allowing the identification of 22 men with bi-allelic mutations in *DNAH1* ($n = 6$), *CFAP43* ($n = 10$), and *CFAP44* ($n = 6$). CRISPR/Cas9 created homozygous *CFAP43/44* male mice that were infertile and presented severe flagellar defects confirming the human genetic results. Immunoelectron and stimulated-emission-depletion microscopy performed on CFAP43 and CFAP44 orthologs in *Trypanosoma brucei* evidenced that both proteins are located between the doublet microtubules 5 and 6 and the paraflagellar rod. Overall, we demonstrate that CFAP43 and CFAP44 have a similar structure with a unique axonemal localization and are necessary to produce functional flagella in species ranging from *Trypanosoma* to human.

[#]A full list of authors and their affliations appears at the end of the paper.

Medical treatment of infertility has rapidly evolved over the past four decades, but much remains to be accomplished[1]. Despite recent success in identifying infertility genes[2–4], most genetic causes of male infertility are currently uncharacterized and additional efforts should be pursued to better characterize male infertility. We demonstrated previously that mutations in the *DNAH1* gene are responsible for multiple morphological abnormalities of the flagella (MMAF), an infertility phenotype characterized by severe asthenozoospermia due to a combination of flagellar defects including short, curled, abnormal width, rolled, or absent flagella[5,6]. *DNAH1* encodes an axonemal inner dynein arm heavy chain, the lack of which leads to a strong disorganization of the axoneme[5]. *DNAH1* mutations were identified in approximately one-third of the studied patients, indicating that MMAF is genetically heterogeneous and that other genes are likely to be involved in this syndrome[2].

In the present study, we analyzed 78 MMAF patients using whole-exome sequencing (WES) and showed that in addition to mutations in *DNAH1*, mutations in *CFAP43* and *CFAP44*, two genes encoding for WD repeat domains (WDR) containing proteins, are responsible for MMAF syndrome and account for 20.5% of this cohort. Most importantly, we investigated the role of these novel genes by performing gene invalidation and silencing in two evolutionary distant models, *Trypanosoma* and mouse, yet sharing an extremely conserved flagellar structure. Using this original approach, we demonstrate the importance of WDR proteins for axonemal structure of flagella and male fertility in humans.

## Results

**Identification of *CFAP43* and *CFAP44* mutations in MMAF patients**. In the present study, we analyzed a cohort of 78 individuals presenting with a MMAF phenotype defined by the presence in the ejaculate of immotile spermatozoa with several abnormalities of the sperm flagellum including short, coiled, absent, and flagella of irregular caliber[5] (Fig. 1a). A majority of patients originated from North Africa, 46 were recruited in Tunisia, 10 in Iran, and 22 in France. The average semen parameters of all 78 MMAF patients included in the cohort are

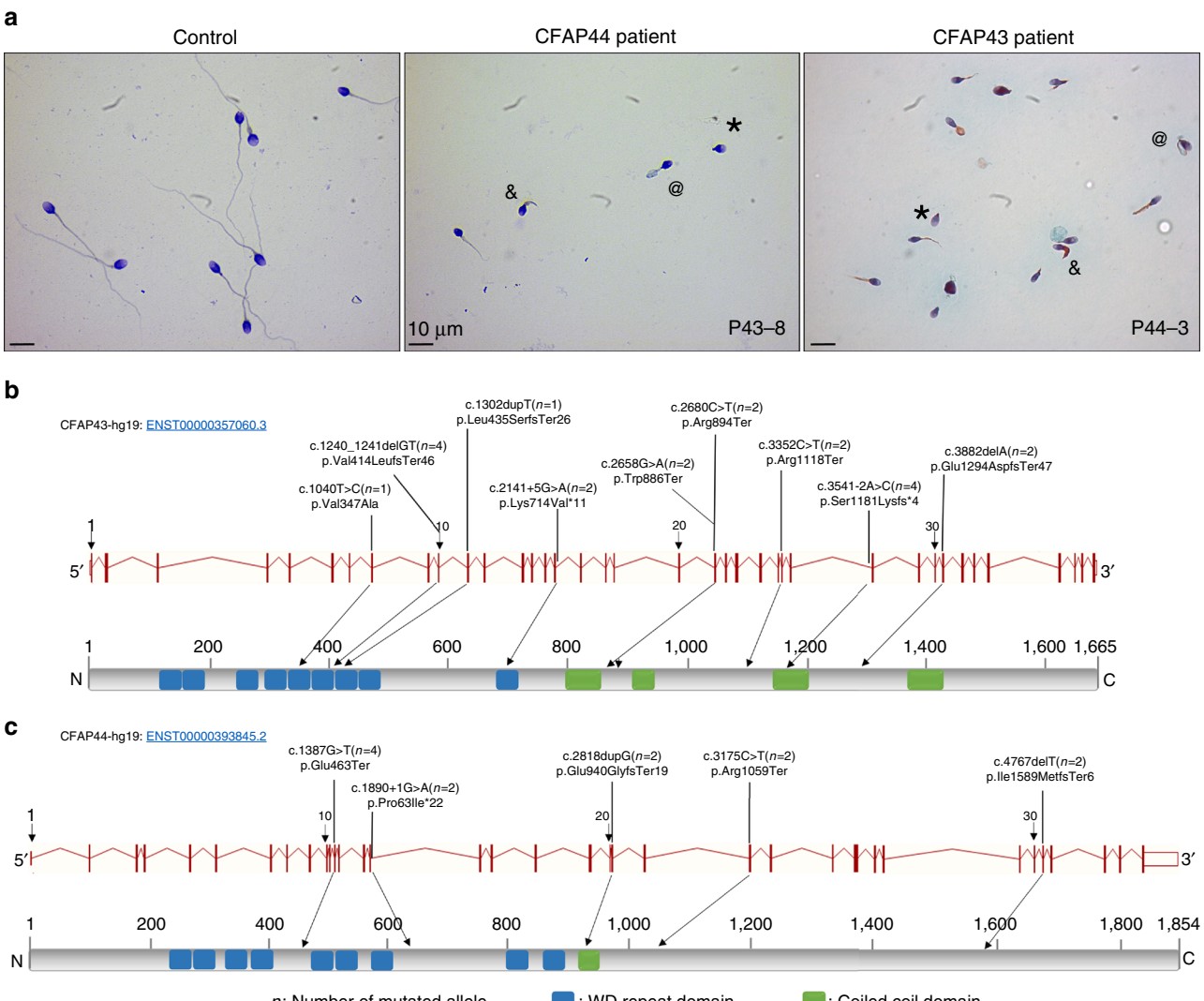

**Fig. 1** Morphology of spermatozoa from the patients $P_{43}$-8 and $P_{44}$-3 and position of *CFAP43* and *CFAP44* mutations in the intron–exon structure and in the protein. **a** All spermatozoa have a much shorter flagellum than those of controls. Additional features of MMAF spermatozoa are absent (*), thick (&), and rolled flagella (@). **b** Mutations identified in the *CFAP43* gene. **c** Mutations identified in the *CFAP44* gene. Blue squares stand for WD repeat domains and green squares for coiled-coiled domains as described by Uniprot server. Mutations are annotated in accordance to the HGVS's recommendations

**Table 1 Average semen parameters in the different genotype groups for the 78 MMAF subjects**

| Semen parameters | MMAF *CFAP43* patients, $n = 10$ | MMAF *CFAP44* patients, $n = 6$ | MMAF *DNAH1* patients, $n = 6$ | MMAF all mutations, $n = 22$ | MMAF with unknown causes, $n = 56$ | Overall MMAF, $n = 78$ |
|---|---|---|---|---|---|---|
| Mean age (years) | $37.7 \pm 9.6$ ($n' = 9$) | $41.3 \pm 4.3$ ($n' = 6$) | $41.5 \pm 4.4$ ($n' = 6$) | $39.8 \pm 7$ ($n' = 21$) | $42.3 \pm 7.9$ ($n' = 56$) | $41.6 \pm 7.7$ ($n' = 77$) |
| Sperm volume (ml) | $3.5 \pm 1.4$ ($n' = 8$) | $3.2 \pm 0.87$ ($n' = 6$) | $3.5 \pm 1.2$ ($n' = 6$) | $3.4 \pm 1.2$ ($n' = 20$) | $3.5 \pm 1.5$ ($n' = 55$) | $3.5 \pm 1.4$ ($n' = 75$) |
| Sperm concentration ($10^6$/ml) | $27.2 \pm 23.4$ ($n' = 8$) | $7.9 \pm 8.4$ ($n' = 6$) | $22.9 \pm 15.2$ ($n' = 6$) | $20.1 \pm 18.8$ ($n' = 20$) | $27.6 \pm 35.7$ ($n' = 55$) | $25.6 \pm 32.1$ ($n' = 75$) |
| Motility (a + b + c) 1 h | $0 \pm 0$ ($n' = 9$) | $0 \pm 0$ ($n' = 6$) | $2.6 \pm 4.2$ ($n' = 6$) | $0.7 \pm 2.4^*$ ($n' = 21$) | $5 \pm 6.1$ ($n' = 55$) | $3.9 \pm 5.6$ ($n' = 76$) |
| Vitality | $55.5 \pm 24$ ($n' = 8$) | $43.3 \pm 22.6$ ($n' = 6$) | $51.2 \pm 23.1$ ($n' = 5$) | $50.5 \pm 22.7$ ($n' = 19$) | $53.4 \pm 20$ ($n' = 53$) | $52.7 \pm 20$ ($n' = 72$) |
| Normal spermatozoa | $1.25 \pm 3.5$ ($n' = 8$) | $0 \pm 0$ ($n' = 6$) | $0 \pm 0$ ($n' = 6$) | $0.5 \pm 2.3$ ($n' = 20$) | $2.1 \pm 4.1$ ($n' = 54$) | $1.6 \pm 2.7$ ($n' = 61$) |
| Absent flagella | $21.8 \pm 17.6$ ($n' = 5$) | $36.8 \pm 4.1$ ($n' = 5$) | $25.6 \pm 15.9$ ($n' = 5$) | $28.1 \pm 14.4^*$ ($n' = 15$) | $18.5 \pm 15.5$ ($n' = 51$) | $20.7 \pm 15.7$ ($n' = 66$) |
| Short flagella | $65.3 \pm 31.7$ ($n' = 8$) | $52.2 \pm 27$ ($n' = 6$) | $49.8 \pm 24.3$ ($n' = 5$) | $57.1 \pm 27.9^*$ ($n' = 19$) | $38.9 \pm 25.7$ ($n' = 53$) | $43.7 \pm 27.3$ ($n' = 72$) |
| Coiled flagella | $8.2 \pm 6$ ($n' = 6$) | $14.4 \pm 7$ ($n' = 5$) | $9 \pm 6.3$ ($n' = 5$) | $10.4 \pm 6.6$ ($n' = 16$) | $13.5 \pm 10$ ($n' = 53$) | $12.8 \pm 9.4$ ($n' = 69$) |
| Bent flagella | $10.3 \pm 6$ ($n' = 3$) | $9$ ($n' = 1$) | $6 \pm 8.5$ ($n' = 2$) | $8.7 \pm 5.8$ ($n' = 6$) | $12 \pm 9$ ($n' = 20$) | $4.2 \pm 8.4$ ($n' = 26$) |
| Flagella of irregular caliber | $20.2 \pm 19.3$ ($n' = 5$) | $28.4 \pm 16.9$ ($n' = 5$) | $35 \pm 22.7$ ($n' = 5$) | $27.9 \pm 19.4$ ($n' = 15$) | $32.8 \pm 26.7$ ($n' = 52$) | $31.7 \pm 25.1$ ($n' = 67$) |
| Multiple anomalies index | $2.3 \pm 0.2$ ($n' = 4$) | $3.4 \pm 0.4$ ($n' = 5$) | $2.3 \pm 1.3$ ($n' = 5$) | $2.7 \pm 1$ ($n' = 14$) | $2.7 \pm 0.6$ ($n' = 47$) | $2.7 \pm 0.7$ ($n' = 61$) |

Values are expressed in percent, unless specified otherwise
Values are mean ± SD; $n$ = total number of patients in each group; $n'$ = number of patients used to calculate the average based on available data
We compared statistical differences between MMAF due to *CFAP43*, *CFAP44*, and *DNAH1* mutations versus MMAF due to uncharacterized genetic cause
CFAP cilia and flagella associated protein; MMAF, multiple morphological abnormalities of the flagella
*A significant difference $P < 0.05$

**Table 2 *CFAP43* (*WDR96*), *CFAP44* (*WDR52*), and *DNAH1* variants identified by WES for all the analyzed subjects ($n = 78$)**

| Gene | Variant coordinates | Transcript | cDNA variation | Amino-acid variation | Patients | Nationality | Hom./Het. |
|---|---|---|---|---|---|---|---|
| *CFAP43* | Chr10:105,912,486 | ENST00000357060 | c.3541−2A>C | p.Ser1181Lysfs*4 | $P_{43}$-1, $P_{43}$-2 | Tunisia | Homozygous |
| *CFAP43* | Chr10:105,956,662 | ENST00000357060 | c.1240_1241delGT | p.Val414LeufsTer46 | $P_{43}$-3, $P_{43}$-4 | Afghanistan, Iran | Homozygous |
| *CFAP43* | Chr10:105,928,535 | ENST00000357060 | c.2658G>A | p.Trp886Ter | $P_{43}$-5 | Algeria | Homozygous |
| *CFAP43* | Chr10:105,928,513 | ENST00000357060 | c.2680C>T | p.Arg894Ter | $P_{43}$-6 | Algeria | Homozygous |
| *CFAP43* | Chr10:105,905,296 | ENST00000357060 | c.3882delA | p.Glu1294AspfsTer47 | $P_{43}$-7 | Tunisia | Homozygous |
| *CFAP43* | Chr10:105,921,781 | ENST00000357060 | c.3352C>T | p.Arg1118Ter | $P_{43}$-8 | Tunisia | Homozygous |
| *CFAP43* | Chr10:105,953,765 | ENST00000357060 | c.1302dupT | p.Leu435SerfsTer26 | $P_{43}$-9 | France | Heterozygous |
| *CFAP43* | Chr10:105,963,485 | ENST00000357060 | c.1040T>C | p.Val347Ala | $P_{43}$-9 | France | Heterozygous |
| *CFAP43* | Chr10:105,944,769 | ENST00000357060 | c.2141+5G>A | p.Lys714Val*11 | $P_{43}$-10 | Turkey | Homozygous |
| *CFAP44* | Chr3:113,114,596 | ENST00000393845 | c.1890+1G>A | p.Pro631Ile*22 | $P_{44}$-1 | Tunisia | Homozygous |
| *CFAP44* | Chr3:113,063,450 | ENST00000393845 | c.3175C>T | p.Arg1059Ter | $P_{44}$-2 | Tunisia | Homozygous |
| *CFAP44* | Chr3:113,082,107_11,3082,108 | ENST00000393845 | c.2818dupG | p.Glu940GlyfsTer19 | $P_{44}$-3 | Morocco | Homozygous |
| *CFAP44* | Chr3:113,119,479 | ENST00000393845 | c.1387G>T | p.Glu463Ter | $P_{44}$-4, $P_{44}$-5 | Algeria | Homozygous |
| *CFAP44* | Chr3:113,023,990 | ENST00000393845 | c.4767delT | p.Ile1589MetfsTer6 | $P_{44}$-6 | Algeria | Homozygous |
| *DNAH1* | Chr3:52,414,073 | ENST00000420323 | c.7531delC | p.Gln2511SerfsTer27 | $P_{DNA}$-1 | Tunisia | Homozygous |
| *DNAH1* | Chr3:52,382,924 | ENST00000420323 | c.2127dupC | p.Ile710HisfsTer4 | $P_{DNA}$-2 | Tunisia | Homozygous |
| *DNAH1* | Chr3:52,395,227 | ENST00000420323 | c.4744_4752delCCAGCTGGC | p.Pro1582_Gly1584del | $P_{DNA}$-3 | Tunisia | Homozygous |
| *DNAH1* | Chr3:52,394,055 | ENST00000420323 | c.4531G>A | p.Val1511Met | $P_{DNA}$-4 | Tunisia | Heterozygous |
| *DNAH1* | Chr3:52,394,397 | ENST00000420323 | c.4642C>G | p.Leu1548Val | $P_{DNA}$-4 | Iran | Heterozygous |
| *DNAH1* | Chr3:52,409,423 | ENST00000420323 | c.7153T>A | p.Trp2385Arg | $P_{DNA}$-5 | Iran | Heterozygous |
| *DNAH1* | Chr3:52,423,486 | ENST00000420323 | c.9505C>G | p.Arg3169Gly | $P_{DNA}$-5 | France | Heterozygous |

cDNA, complementary DNA; CFAP cilia and flagella associated protein; WES, whole-exome sequencing

described in Table 1. Nearly no spermatozoa with normal morphology could be observed in the ejaculate of MMAF individuals (1.6%); an average of 20.7 and 43.7% of spermatozoa had no flagella and short flagella, respectively, and 31.7% of the spermatozoa had flagella with an irregular caliber. As a result, total sperm motility was dramatically reduced to 3.9% (normal value >40%), which prevented natural conception for all individuals. Given the notion of consanguinity for most individuals from the cohort, we postulated that infertility was likely transmitted through recessive inheritance and probably often resulted from homozygous mutations. After exclusion of frequent variants and applying stringent filters, a limited list of homozygous variants was identified for each proband. First, we identified six patients (7.7%) with mutations in the *DNAH1* gene (Table 2), previously identified as the main cause of MMAF phenotype[5,6]. We subsequently identified 10 subjects with variants in *CFAP43* (12.8%), eight of which had a homozygous loss-of-function variant and two with two likely deleterious variants (Table 2). In addition, six subjects (7.7%) had a homozygous loss-of-function variant in *CFAP44* (Table 2). These two *CFAP* genes (for cilia and flagella associated protein) were reported in public expression databases to be strongly expressed in the testis and to be connected with cilia and flagella structure and/or functions[7]. Quantitative real-time reverse transcription PCR (RT-qPCR) experiments performed in human and mouse tissue panels confirmed that *CFAP43* and *CFAP44* mRNA in testis was predominant and very significantly higher than in the other tested tissues (Supplementary Fig. 1). Taking into account the high number of mutated patients and the specific expression pattern of the two genes, we focused on these two genes, which appeared as

the best candidates to explain the primary infertility observed for these individuals. All individuals with *CFAP43* or *CFAP44* mutations were unrelated; to our knowledge, none carried rare variants in genes previously reported to be associated with male infertility. *CFAP43* and *CFAP44* encoded proteins that belong to the WDR protein family and are both composed of nine WD repeats[8].

*CFAP43* (also known as *WDR96*, NM_025145) is localized on chromosome 10 and contains 38 exons encoding a predicted 1,665-amino-acid protein (Q8NDM7). We identified 9 different pathogenic variants in *CFAP43* in 10 unrelated individuals from the cohort. The splicing variant c.3541−2A>C was identified in two unrelated individuals and affects a consensus splice acceptor site of *CFAP43* intron 27 (Table 2, Fig. 1b). This variant was absent from the 60,706 unrelated individuals aggregated in the ExAC database (http://exac.broadinstitute.org), consistent with a deleterious effect on protein function. Three other variants correspond to stop-gain mutations identified in three subjects: c.2658G>A and c.2680C>T are located in exon 21 and c.3352C>T in the exon 26 (Fig. 1b). These three variants were found in the ExAC database with very low allele frequency ranging from $8.24e^{-06}$ to $9.89e^{-05}$. Two others mutations were small frameshift indels not listed in ExAC: c.1240_1241delGT (found in two unrelated patients) and c.3882delA (Table 2, Fig. 1b). All these mutations generate premature stop codons and are predicted to produce no protein or truncated non-functional proteins. Apart from these obvious harmful mutations found in eight patients, we also identified two patients harboring likely deleterious mutations in *CFAP43*. The first patient harbors a homozygous splicing mutation c.2141+5G>A, not listed in ExAC which, according to the splice site prediction algorithm Human Splicing Finder (http://www.umd.be/HSF3), likely alters the consensus splice donor site of *CFAP43* exon 16. Unfortunately, we could not obtain any additional biological material from this patient and could not verify the effect of this variant on mRNA expression. The second patient is a compound heterozygous carrying one single-nucleotide duplication, c.1320dupT, located in the exon 11 and an additional missense mutation, Val347Ala, located in exon 8 of the *CFAP43* gene. Although prediction software classified this last variant as likely "benign," this mutation affects a conserved residue located in the N-terminal part of the protein within a WDR known to be important for protein/protein interactions[9]. Moreover, this missense variant is found at a very low prevalence in the general population estimated to 7.45e−5 (9/12078). Altogether the above-identified mutations clearly indicate that in humans, loss of function of *CFAP43* is associated with asthenozoospermia and sperm flagellum defects.

*CFAP44* (also known as WDR52, NM_001164496) is localized on chromosome 3 and contains 35 exons encoding a predicted 1,854-amino-acid protein (Q96MT7). Five different homozygous variants were identified in six unrelated patients (Table 2, Fig. 1c). Stop-gain mutations c.1387G>T and c.3175C>T are located in exons 12 and 23, respectively. The frameshift mutations, c.4767delT, with a single-nucleotide deletion, are located in exon 31, and the variant c.2818dupG, with a single-nucleotide duplication, is located in exon 21. The last mutation, c.1890 +1C>T, affects exon 15 consensus splice donor site. All these mutations are predicted to generate a premature stop codon. None of these variants were listed in the ExAC database.

The presence of all variants was verified by bidirectional Sanger sequencing (Supplementary Fig. 2). The detailed sperm parameters of the patients carrying *DNAH1*, *CFAP43*, or *CFAP44* mutations were compared with each other. There was no significant difference between the three groups (Table 1) as illustrated by the similar morphologies of sperm from patient

$P_{43}$-8 and $P_{44}$-3, mutated for *CFAP43* and *CFAP44*, respectively (Fig. 1a). Data from all these groups were therefore pooled and compared with data from the patients with no identified mutation. Patients with a mutation had a significantly higher rate of spermatozoa with short or absent flagella and presented a significantly lower motility rate compared to patients with no identified mutations. There was no difference in the other sperm parameters (Table 1).

**CFAP43 and CFAP44 mutations induce severe axonemal disorganization.** The internal skeleton of motile cilia and flagella, named the axoneme, is a highly evolutionary conserved structure, which consists of nine doublets of microtubules (DMTs) circularly arranged around a central pair complex (CPC) of microtubules ('9+2' structure). All DMTs are not identical and are organized in a highly reproducible fashion with substructures specific to certain doublets. The convention adopted in the numbering of the DMTs was based on the relative position of each DMT with regard to the plane of the CPC[10,11]. The beating of cilia and flagella is orchestrated by multiprotein-ATPase complexes, located on the peripheral doublets, which provide the sliding force for sperm motility. In addition, the sperm flagellum harbors specific peri-axonemal structures, which are not found in other motile cilia, a helical mitochondrial sheath (MS) in the midpiece, the fibrous sheath (FS) in the principal piece (PP), and outer dense fibers (ODFs) in the midpiece and the proximal part of the PP.

We studied the ultrastructure of sperm cells from mutated patients in *CFAP43* and *CFAP44* by transmission electron microscopy (TEM) (Fig. 2). Due to insufficient amount of sperm cells collected from most *CFAP44* and *CFAP43* patients, only one patient for each gene was studied by TEM. For each patient, we could observe longitudinal sections and >20 cross-sections presenting a sufficient quality to observe the ultrastructure of the axonemal components. Longitudinal sections showed severe axonemal and peri-axonemal defects affecting the ODF, the FS, and the MS, which appeared completely disorganized resulting in aborted flagella or their replacement by a cytoplasmic mass englobing unassembled axonemal components (Fig. 2). In *CFAP43*-mutated and *CFAP44*-mutated patients, approximately 95% of the cross-sections were abnormal and the main defect observed was the absence of the CPC (9+0 conformation) observed in 81.8, and 66.7% of the cases, respectively, compared to 0% in control fertile subject (Supplementary Table 1). In the residual fraction (~5%) of axoneme with normal (9+2) conformation, peri-axonemal structures abnormalities were constantly observed (Supplementary Table 1). The lack of central pair defects was associated with peripheral doublets disorganization in 13.6 and 19% for *CFAP43* and *CFA44*, respectively (Supplementary Table 1). Cross-sections with a single central microtubule (9+1 conformation) were observed in about 10% of cases, only for the *CFAP44* patient. Interestingly, in CFAP43 patients, the CPC, when present, was misoriented compared to control sections in which the CPC is normally parallel to the axis of the two longitudinal columns of the FS (Fig. 2).

To define the ultrastructural defects evidenced by TEM, we performed immunofluorescence (IF) experiments using antibodies targeting different axonemal proteins. In *CFAP43*-mutated and *CFAP44*-mutated patients, staining of SPAG6, a protein located in the CPC, was abnormal and atypical compared to control staining (Fig. 3a–c). SPAG6 staining was absent in CFAP43 patient sperm cells, whereas in *CFAP44* patients' sperm cells, SPAG6 immunostaining was present but displayed an abnormal and diffuse pattern concentrated in the midpiece of the spermatozoa, quickly diminishing along the flagellum (Fig. 3d–i).

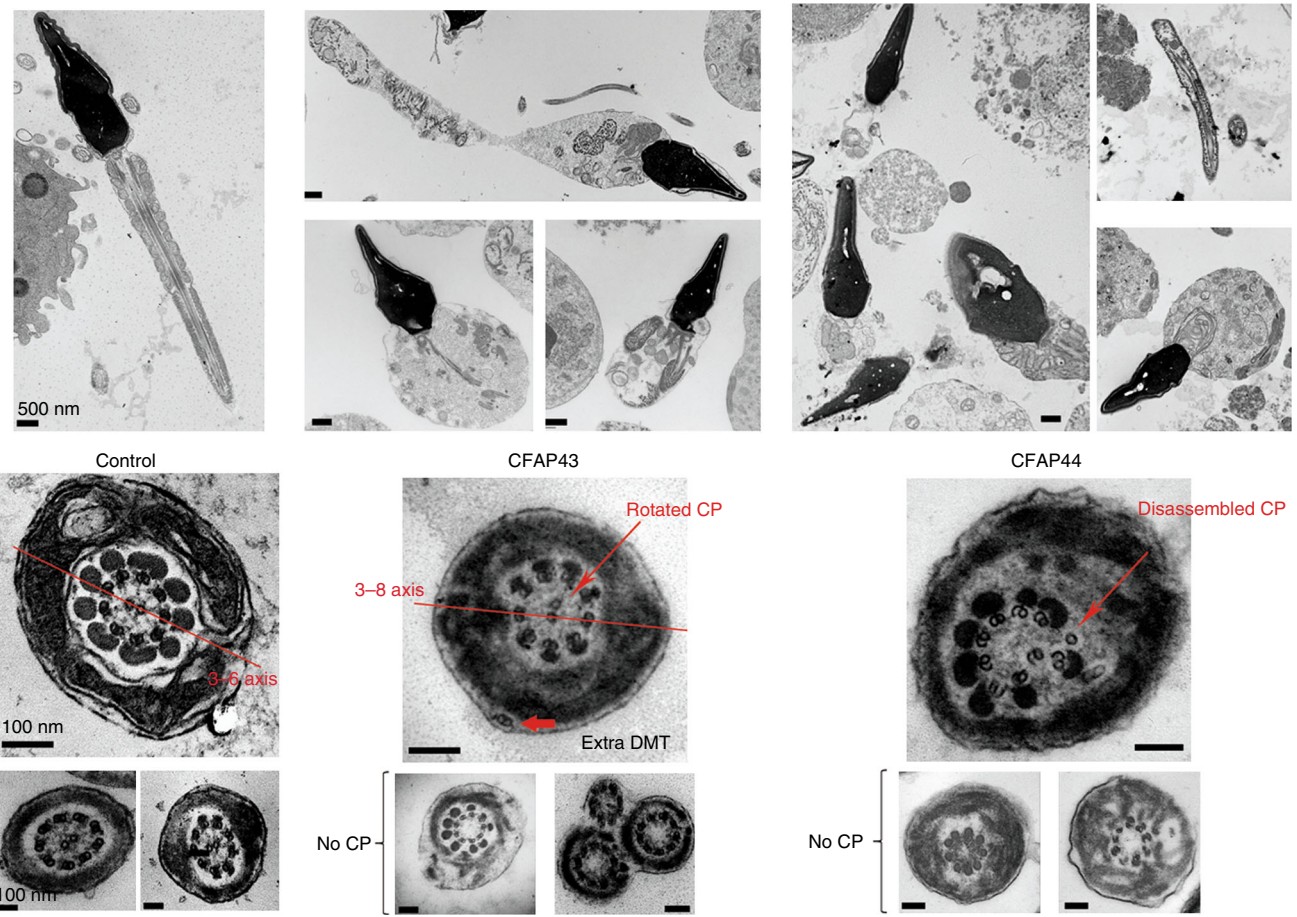

**Fig. 2** Transmission electron microscopy analysis of sperm cells from *CFAP43* and *CFAP44* patients shows a severe axonemal disorganization. (Left panel) Longitudinal sections (scale bar = 500 nm) and cross-sections (scale bars, 100 nm) of sperm flagellum from control. (Central panel) Longitudinal sections (scale bars, 500 nm) and cross-sections (scale bars, 100 nm) of sperm flagellum from *CFAP43*-mutated patient (P$_{43}$-8). We can notice a short short tail corresponding to a cytoplasmic mass containing the different components of the flagellum, all unorganized. In CFAP43 upper cross-section, the CPC is not aligned with DMTs 3 and 8 (red line) and is rotated by 90°. We can observe the absence of central pair of microtubules in other cross-sections. (Right panel) Ultrastructure of *CFAP44*-mutated sperm (P$_{44}$-3) longitudinal sections (scale bars = 500 nm) show similar ultrastructure of short tail (cytoplasmic mass). In CFAP44 upper cross-section, the central pair is disassembled and displaced (red arrow). We can observe the absence of central pair of microtubules in other cross-sections. Scales bars for cross-sections = 100 nm

Additionally, staining of the radial spoke head protein RSPH1 presented significantly different patterns compared to controls. In *CFAP43*-mutated patients, the RSPH1 staining was completely abnormal with a marked diffuse staining, concentrated in the midpiece, whereas the tubulin staining remained restricted to the axoneme (Fig. 3m–o). In *CFAP44*-mutated patients' sperm cells, RSPH1 staining was significantly reduced (Fig. 3p–r). For *CFAP43*-mutated and *CFAP44*-mutated patients, immunostaining for AKAP4, DNALI1, DNAI2, and GAS8 were all comparable with controls (Fig. 3j–l), suggesting that FS, outer dynein arms, inner dynein arms, and the nexin-dynein regulatory complex, respectively, were not directly affected by mutations in *CFAP43* or *CFAP44* (Supplementary Table 2—Human). Unfortunately, we could not obtain any specific antibodies against CFAP43 and CFAP44 that would allow the localization of these proteins in mouse and human flagella.

**Analysis of *Cfap43* and *Cfap44* knockout mice.** We assessed the impact of *Cfap43* and *Cfap44* absence on mouse spermatogenesis by generating knockout (KO) animals using the CRISPR-Cas9 technology. We obtained four independent mutational events for

*Cfap43* and three for *Cfap44*; all were insertions/deletions of a few nucleotides inducing a translational frameshift expected to lead to the complete absence of the protein or the production of a truncated protein. Because all KO strains for *Cfap43* presented the same reproductive phenotype (identical sperm morphology associated with complete infertility), we randomly chose one and restricted our study to a strain with a 4 bp deletion in exon 21 (delAAGG). Similarly, the *Cfap44* lines presented the same reproductive phenotype, and we focused on a strain with a 7 bp insertion in exon 3 (InsTCAGATA). RT-PCR was performed on testis RNA from *Cfap43*$^{-/-}$ and *Cfap44*$^{-/-}$ mice, which confirmed the production of abnormal transcripts in both mutants leading to a premature stop codon (Supplementary Fig. 3).

The reproductive phenotype was studied for both KO mice models. Homozygous KO females were fully fertile and gave litters of normal size (7.8 ± 1.8 and 7.3 ± 3.5 versus 6.7 ± 0.5 pups/litter (mean ± SD, $n = 7$) for *Cfap43*$^{-/-}$, *Cfap44*$^{-/-}$, and wild-type (WT), respectively), contrary to homozygous KO males, which exhibited complete infertility when mated with WT females (Fig. 4a). Epididymal sperm concentrations were 11.4 ± 0.9, 11.2 ± 0.4, and 14.4 ± 3.1×10$^6$ sperm/ml (mean ± SD, $n = 3$), for *Cfap43*$^{-/-}$, *Cfap44*$^{-/-}$, and WT, respectively, and fall

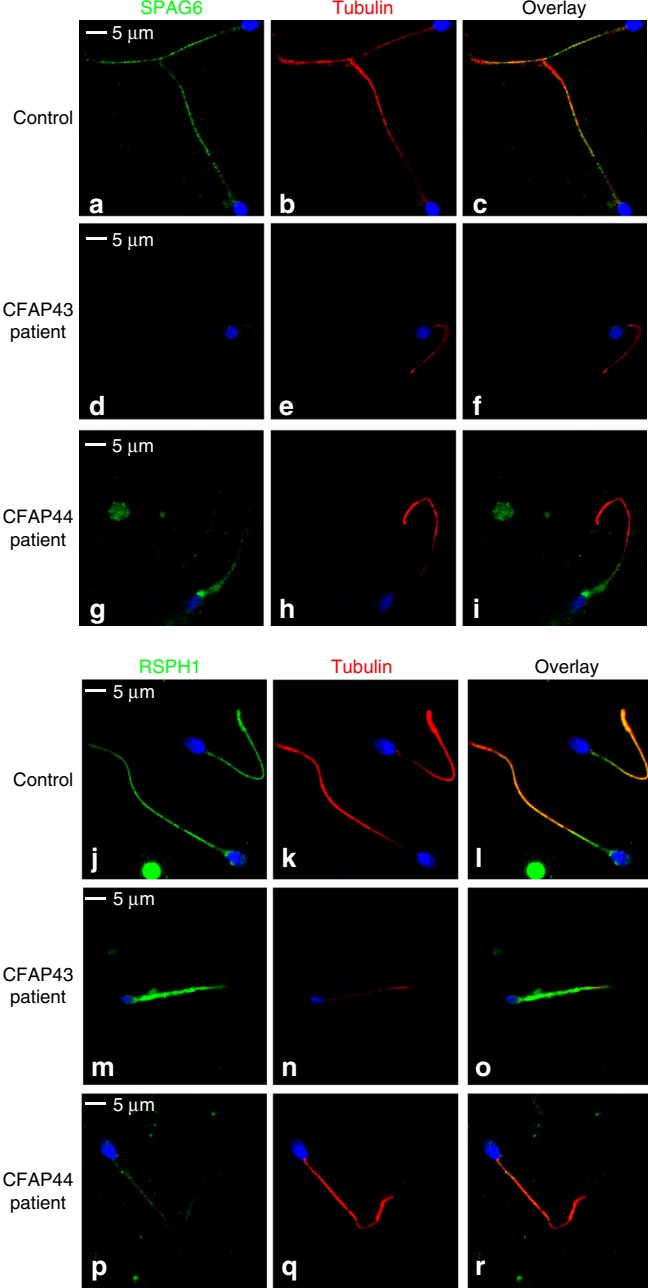

**Fig. 3** Immunofluorescence staining in *CFAP43* and *CFAP44* patients reveals an abnormal axonemal organization. **a–c** sperm cells from a fertile control stained with anti SPAG6 (green), a protein located in the CPC, and anti-acetylated tubulin (red) antibodies. DNA was counterstained with Hoechst 33342. **c** Corresponds to **a** and **b** overlay and shows that in control sperm, SPAG6 and tubulin staining superimpose. Scale bars = 5 μm. **d–f** SPAG6 staining is absent in sperm from the patient P$_{43}$-5 homozygous for the c.2658C>T variant in *CFAP43*. **d–i** Similar IF experiments performed with sperm cells from the patient P$_{44}$-2 homozygous for the c.3175C>T variant in *CFAP44*. Scale bar = 5 μm. Contrary to the control, the SPAG6 immunostaining (green) is abnormal with a diffuse pattern concentrated in the midpiece of the spermatozoa and is not detectable in the principle piece. **j–l** Sperm cells from a fertile control stained with anti RSPH1 (green), a protein of the radial spoke's head, and anti-acetylated tubulin (red) antibodies. DNA was counterstained with Hoechst 33342. **l** corresponds to **j** and **k** overlay and shows that RSPH1 and tubulin staining superimpose in control sperm. Scale bar = 5 μm. **m–o** In sperm from the patient P$_{43}$-5, RSPH1 staining (green) is significantly different from control (**m**) with a marked diffuse staining. **p–r** In sperm from the patient P$_{44}$-2 the intensity of the RSPH1 staining is strongly reduced

within the normal values for the mouse. We next studied sperm morphology and observed that for *Cfap43*$^{-/-}$ males, 100% of sperm displayed a typical human MMAF phenotype with short, thick, and coiled flagella (Fig. 4b); although slightly deformed, sperm head exhibited an overall normal hooked form (Fig. 4e). In contrast, sperm from *Cfap44*$^{-/-}$ males had normal flagellum length, but most of them showed abnormal forms and irregular caliber of the midpiece (Fig. 4c–f). In sperm from *Cfap44*$^{-/-}$ mice, staining of Mpc1l, a sperm mitochondrial protein[12], was discontinuous or punctiform, strongly suggesting that the distribution of the mitochondria surrounding the axoneme was irregular, consistent with the observed midpiece abnormalities (Fig. 4h–m). For both *Cfap43*$^{-/-}$ and *Cfap44*$^{-/-}$ sperm, the observed structural flagellum defects were associated with severe motility deficiencies (Fig. 4g) with sperm from *Cfap43*$^{-/-}$ males showing complete immotility whereas those from *Cfap44*$^{-/-}$ sometime presented some small vibrations (Supplementary Movie 1), in contrast to the motile sperm from *Cfap44*$^{+/-}$ heterozygotes (Supplementary Movie 2). Interestingly, we observed that in heterozygous animals, *Cfap43*$^{+/-}$ mice had an impaired progressive sperm motility and *Cfap44*$^{+/-}$ mice had an increased proportion or morphologically abnormal spermatozoa (Fig. 4g).

To better characterize the molecular defects induced by the absence of Cfap43 and Cfap44 in mouse sperm, we studied by IF the presence and localization of several proteins belonging to different substructures of the axoneme in sperm from both KO mouse models. The presence of the following proteins was investigated: Dnah5 and Dnali1 as markers of dynein arms, Rsph1 and Rsph4a as markers of radial spokes, Gas8 as a marker of nexin links, and Spef2 as a marker of the central pair (Supplementary Table 2—Mouse). In *Cfap43*$^{-/-}$-mutated animals, Spef2, Rsph1, and Rsph4a were clearly missing (Fig. 5d–f, j–l) as compared to controls (Fig. 5a–c, g–i, Supplementary Table 2, and Supplementary Fig. 4), indicating that the center of the axoneme, including the head of the radial spoke interacting with the central pair, was absent. In contrast, no obvious defects were observed in sperm from *Cfap44*$^{-/-}$, suggesting that all substructures were present (Supplementary Table 2—Mouse).

The impact of the absence of these proteins on the flagellum ultrastructure was analyzed on sperms from *Cfap43*$^{-/-}$ and *Cfap44*$^{-/-}$ by TEM. In the midpiece of WT mouse flagellum, the 9+2 axoneme is surrounded by nine ODFs and the mitochondria sheath[13]. The ODFs 1, 5, 6, and 9 are parallel and aligned with the central pair (Fig. 5m, see also Fig. 6a and Supplementary Fig. 5a for the DMT annotation). In addition, a transversal complex is composed of the central pair and DMTs 3 and 8. This complex includes ODFs 3 and 8 in the midpiece and is closely associated with the FS in the PP through two stalks emerging from the longitudinal column and linking DMTs 3 and 8[14]. In the PP, the axoneme is surrounded by seven ODFs (3 and 8 are missing) and by an FS containing two longitudinal columns. Contrary to what is usually observed in human sperm, this organization is highly reproducible and <5% of WT sections from mouse sperm flagellum present structural defects (Supplementary Fig. 6a). Analyses of longitudinal sections from *Cfap43*$^{-/-}$ sperm showed that the observed short tail actually corresponds to a large cytoplasmic mass containing unorganized structural components of the flagellum (Supplementary Fig. 7). When the axoneme was present, all transversal sections observed revealed a substantial structural disorganization with uneven distribution of the nine DMTs and the absence of the CPC (Fig. 5n). In contrast, in *Cfap44*$^{-/-}$ sperm, the 9+2 organization of the axoneme was preserved in 70% of cases (Fig. 6b, d, e, g–i). The observed defects included the absence of peripheral doublets, external shift, or mislocalization of the central pair and distorted circular

distribution of the DMTs with CPC misorientation (Fig. 6d–f). *Cfap44*$^{-/-}$ sperm also exhibited important defects of the ODFs and of the FS (Supplementary Fig. 6b). In the midpiece, the number of dense fibers was significantly higher and their localization and orientation were defective (Fig. 6b, c). In the

PP, ODFs 3 and 8 were abnormally retained, preventing normal anchoring of the FS's stalks of on DMTs 3 and 8 (Fig. 6g–i). The longitudinal columns also presented several defects: they were misaligned and not facing the 3-central-8 complex, leading to notable asymmetry of the structure (Fig. 6k–n), which was

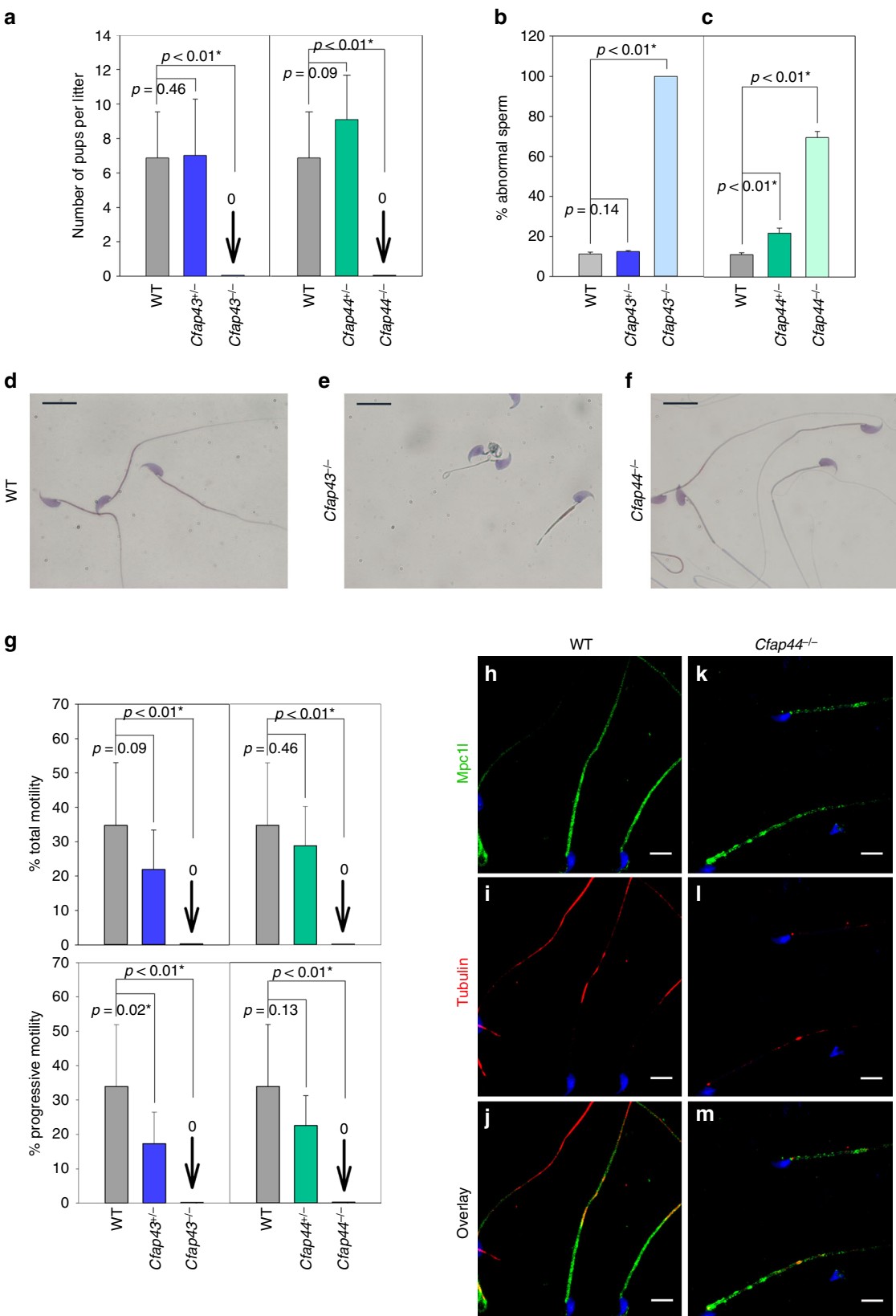

reinforced in numerous sperm by the presence of a third longitudinal column (Fig. 6m, n). Finally, the defects detected in $Cfap44^{-/-}$ MS by optic and fluorescence microscopy were specified by TEM analyses of sperm midpieces, evidencing uneven distribution and fragmentation of the mitochondria (Supplementary Fig. 8).

**TbCFAP43 and TbCFAP44 are necessary axonemal proteins.** To better characterize CFAP43 and CFAP44 localization and function and to overcome the absence of reliable antibodies in human and murine cells, we decided to use *Trypanosoma brucei* (*T. brucei*) as an additional model organism because this flagellated protozoan has largely contributed to elucidating the molecular pathogeny of human ciliopathies[15]. The *T. brucei* axoneme is similar to that of mammalian flagella, but its peri-axonemal structure, although comparable in function, is different. Instead of the FS and the ODFs found in mammals' flagella, it contains the paraflagellar rod (PFR), a complex structure connecting with the axonemal doublets 4–7[16,17] which plays a role in flagellum motility[18,19] and serves as a platform for metabolic and signaling enzymes[20–22]. In *T. brucei* the DMTs are numbered from the doublet opposite the PFR (no. 1), which is normally parallel to the central singlets. The doublets are then numbered clockwise, doublets 5 and 6 facing the PFR and doublets 4 and 7 being positioned on each side of the PFR (Supplementary Fig. 5b for DMT annotation and relative PFR localization).

BLASTp analysis on *T. brucei* genome database[23] using human CFAP43 and CFAP44 sequences identified *T. brucei* orthologs Tb927.7.3560 (named TbCFAP44 in this study) and Tb927.4.5380 (named TbCFAP43 in this study), respectively. The sequence identity, calculated with clustal omega[24], between the four proteins shows that CFAP44 is closest to TbCFAP44 with 25.6% sequence identity and CFAP43 closest to TbCFAP43 with 20.8% sequence identity. Previous functional genomics and proteomic studies identified TbCFAP44 and TbCFAP43 as flagellar proteins[25,26]. In addition, TbCFAP44 is the *Chlamydomonas* FAP44 ortholog, and is involved in flagellar motility (also referred to as *T. brucei* components of motile flagella 7 (TbCMF7 in ref.[27]). However, the function of TbCFAP44 and TbCFAP43 are currently unknown. To confirm that both *Trypanosoma* proteins are the orthologs of the mouse and human proteins, we compared their secondary structure. The secondary structure predicted by Porter 4.0[28] shows that the four proteins share a common general motif with β-strand domains in the first half of the protein and α-helical domains in the second half (C-Ter) (Supplementary Fig. 9). We note that in CFAP43 and TbCFAP43, all residues between amino acids 719 and 1,657 and 631 and 1,446, respectively, are predicted to form α-helices. Interestingly, analyses using Superfamily[29,30] indicated that long α-helices stretches such as those identified in CFAP43 (amino acids 719–1,657) have not yet been reported in any known fold. According to SCOP[31], the N-terminal part of all four proteins (almost half of the proteins) belongs to the class of "All β-proteins" and "7-bladed β-propeller fold." These

N-terminal domains also belong to the WD40 repeat-like superfamily (Supplementary Table 3). This structural comparative analysis indicates that TbCFAP44 and TbCFAP43 and their human orthologs share unique structural similarities and therefore demonstrates that these proteins belong to the same family.

We localized TbCFAP44 and TbCFAP43 in bloodstream from *T. brucei* (BSF, present in the mammalian host), using 10TY1-tagged and 3myc-tagged proteins by generating *T. brucei* cell lines expressing endogenous levels of N-terminal 10TY1-tagged proteins ($_{10TY1}$TbCFAP44 and $_{10TY1}$TbCFAP43) and C-terminal 3myc-tagged proteins (TbCFAP44$_{myc}$ and TbCFAP43$_{myc}$). TY1 tag (EVHTNQDPLDGS, 1.8 kDa) is used as an epitope for immunolabeling[32] used for numerous immunolocalizations in trypanosomes, either as 1xTY1, 3xTY1, or, more recently, 10xTY1[33]. Both proteins, whatever the tag used, were found in the axoneme as substantiated by co-labeling with an antibody against the PFR structure (Supplementary Figs. 10a, b). We acknowledge that the tag might interfere with protein function and localization; however, this result suggests that the 10TY1 tag did not interfere with the trafficking of either proteins which extended throughout the whole flagellum length, as demonstrated by the 3myc and 10TY1 labeling preceding the PFR at the posterior end of the flagellum (next to the kinetoplast) and along the flagellum up to its anterior end. To further investigate this localization we used the 10TY1 tag which produces a strong IF staining[32] necessary for stimulated-emission-depletion (STED) confocal microscopy[34] and created two double-labeled cell lines $_{10TY1}$TbCFAP44/TbCFAP43$_{myc}$ and $_{10TY1}$TbCFAP43/TbCFAP44$_{myc}$. Triple labeling STED confocal microscopy of 10TY1-tagged proteins, tubulin, and PFR2 on permeabilized whole cells demonstrated that $_{10TY1}$TbCFAP44 and $_{10TY1}$TbCFAP43 are closely associated to the axoneme along the PFR (Fig. 7a–f, respectively); similar results were obtained on detergent-extracted cytoskeletons. To confirm this remarkable localization, TEM immunogold labeling was performed using differently tagged proteins. We used the myc tag, a small tag of 10 amino acids which is known to have minimum effect on localization. Concordant with the STED microscopy results, all immunogold beads were located only along the axoneme and facing the PFR, therefore indicating that TbCFAP44 and TbCFAP43 are in direct contact only with the doublets facing the PFR, which correspond to DMTs 5 and 6[35] (Fig. 7g–j). The intervals between two adjacent gold particles were measured and the minimal distance between two particles was around 25 nm for both proteins (Fig. 7g, h, inset). Finally, 10TY1-tagged proteins presented an identical subcellular localization in TEM (Supplementary Fig. 11)

To evaluate the impact of the absence of TbCFAP44 and TbCFAP43 on the structure of the trypanosome flagellum and its beating, we knocked down protein expression by inducible RNA interference (RNAi), either in the parental cell line (cell lines TbCFAP44$^{RNAi}$, TbCFAP43$^{RNAi}$) or in the cell lines expressing the myc-tagged proteins (cell lines TbCFAP44$_{myc}$$^{RNAi}$, TbCFAP43$_{myc}$$^{RNAi}$) (Fig. 8). Efficiency and specificity of RNAi knockdown of TbCFAP44 and TbCFAP43 were validated by

**Fig. 4** Reproductive phenotype of homozygous and heterozygous *Cfap43* and *Cfap44* male mice. **a** Fertility of *Cfap43*$^{+/-}$, *Cfap44*$^{+/-}$ *Cfap43*$^{-/-}$, and *Cfap44*$^{-/-}$ males. Heterozygous and homozygous mutant males were mated with WT females and the numbers of pups per litter were measured. KO males were completely sterile. **b**, **c** Spermatocytograms showing the number of abnormal sperm in heterozygous and homozygous mutant males. **d–f** Images of typical sperm stained with Harris–Shorr from *Cfap43*$^{-/-}$ and *Cfap44*$^{-/-}$ males. Scale bars = 10 µm. **g** Total and progressive motilities of sperm from *Cfap43*$^{-/-}$ and *Cfap44*$^{-/-}$ males. **h–m** The mitochondria sheath is fragmented in *Cfap44*$^{-/-}$ males. Staining of WT sperm with an anti-MCPl1 (**h**, green), a mitochondrial transporter, and anti-acetylated tubulin (**i**, red) antibodies. **j** Overlay of MCPl1 and tubulin staining. Sperms were counterstained with Hoechst (blue). **k–m** Similar experiments on sperm from *Cfap44*$^{-/-}$ males. Scale bars = 10 µm. **a–c**, **g** Data represent mean ± SD; statistical differences were assessed with *t* test, *P* value as indicated

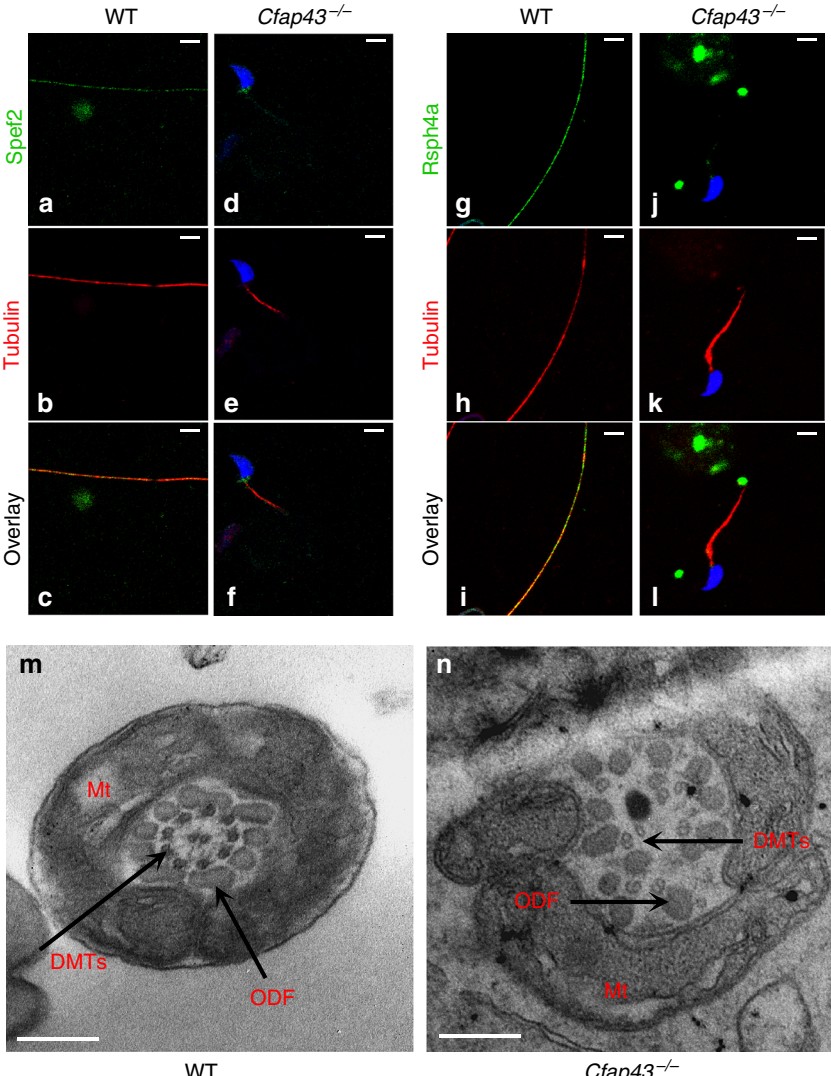

**Fig. 5** The sperm axoneme from *Cfap43*<sup>−/−</sup> males is fully disorganized. **a** Central pair is absent in sperm from *Cfap43*<sup>−/−</sup> males. Staining of WT sperm with an anti-Spef2, a marker of the projection 1b of singlet C1 (**a**, green) and anti-acetylated tubulin (**b**, red) antibodies. **c** Overlay of Spef2 and tubulin staining. Sperms were counterstained with Hoechst (blue). **d–f** Similar experiments on sperm from *Cfap43*<sup>−/−</sup> males. Scale bars = 5 μm. **g–l** Head of radial spoke are absent in sperm from *Cfap43*<sup>−/−</sup> males. Staining of WT sperm with an anti-Rsph4a, another protein of the RS head (**g**, green) and anti-tubulin (**h**, red) antibodies. **i** Overlay of Rsph4a and tubulin staining. Sperm were counterstained with Hoechst (blue). **j–l** Similar experiments on sperm from *Cfap43*<sup>−/−</sup> males. Scale bars = 5 μm. **m**, **n** Transversal section of a sperm from WT (**m**) and *Cfap43*<sup>−/−</sup> (**n**) males observed by EM in the midpiece region. Note the specific arrangement of the ODFs around the axoneme in WT sperm and the complete disorganization of the DMTs and the absence of the central pair in *Cfap43*<sup>−/−</sup> sperm. Scale bars = 240 nm. Number of cross-section for WT and *Cfap43*<sup>−/−</sup> were 21 and 30, respectively. All cross-section were defective for *Cfap43*<sup>−/−</sup>. DMTs, doublet of microtubules; ODF, outer dense fiber; Mt, mitochondria

RT-PCR (Supplementary Fig. 12) and by IF, showing a clear decrease of myc labeling in the new flagellum of *TbCFAP44*<sub>myc</sub><sup>RNAi</sup>-induced and *TbCFAP43*<sub>myc</sub><sup>RNAi</sup>-induced cells (Fig. 8e–l, respectively). Cell proliferation was assessed in parental, non-induced, and induced *TbCFAP44*<sup>RNAi</sup> and *TbCFAP43*<sup>RNAi</sup> cells (and in *TbCFAP44*<sub>myc</sub><sup>RNAi</sup> and *TbCFAP43*<sub>myc</sub><sup>RNAi</sup> cell lines, showing the same results). In all cases, both *TbCFAP44*<sup>RNAi</sup>-induced and *TbCFAP43*<sup>RNAi</sup>-induced cells showed abnormal beating (Supplementary Movies 2–5) and stopped proliferating after 24 h and eventually died (Fig. 8m). These growth defects were accompanied by a defect in cytokinesis producing multi-flagellated cells (Supplementary Fig. 13), a phenotype previously described when proteins directly or indirectly involved in flagellar motility are knocked down in BSF[36]. We observed that the absence of TbCFAP44<sub>myc</sub> and TbCFAP43<sub>myc</sub> did not impact

flagella length (Fig. 8n). Overall, these results indicate that TbCFAP44 and TbCFAP43 are necessary for flagellar function and thus cell proliferation in *T. brucei* BSF.

To assess if the observed cellular phenotypes are caused by each protein independently or if the absence of one induces the absence of the other, we took advantage of the double-labeled cell lines <sub>10TY1</sub>*TbCFAP44/TbCFAP43*<sub>myc</sub> and <sub>10TY1</sub>*TbCFAP43/TbCFAP44*<sub>myc</sub>. Inactivation of TbCFAP43 by RNAi did not alter the amount and localization of TbCFAP44; conversely, inactivation of TbCFAP44 had no impact on TbCFAP43 protein (Supplementary Fig. 14).

The impact of *TbCFAP44* and *TbCFAP43* knockdown was also investigated by TEM. In control longitudinal sections of parental cells, the flagellum exits the cell through the flagellar pocket (Fig. 8o), and, in cross-section, the canonical ultrastructure of the

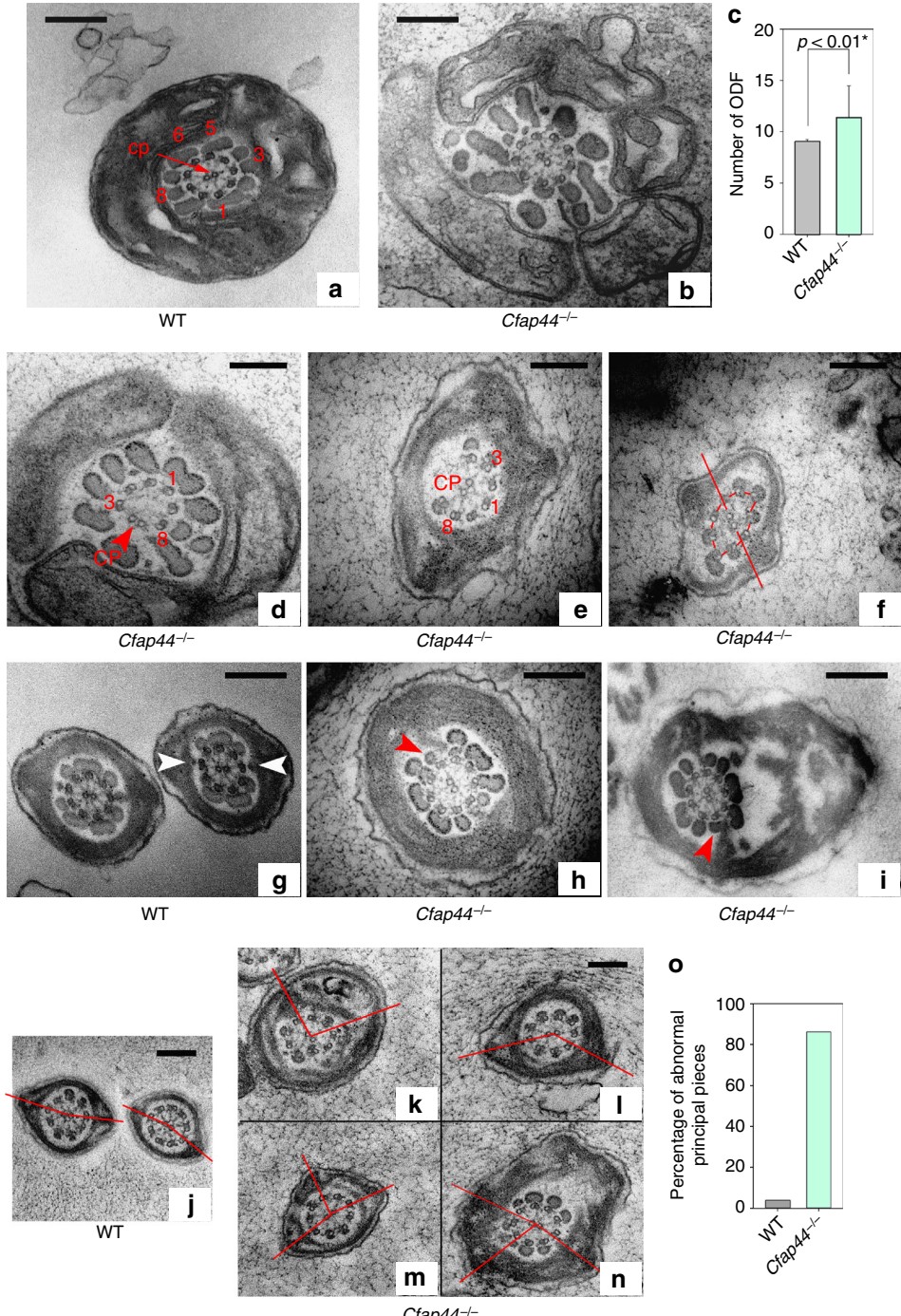

**Fig. 6** Electron microscopy cross-sections of sperm from *Cfap44*$^{-/-}$ males reveal multiple structural axonemal defects. **a** Cross-section of the midpiece of a WT sperm, showing the arrangement of the ODFs around the axoneme. **b** Presence of extra ODFs in midpieces sections of sperm from *Cfap44*$^{-/-}$ males. The orientation of ODFs is also defective, leading to an increase of the midpiece diameter. **a**, **b** Same scale bars = 250 nm. **c** Graph showing the increased number of ODF in the mutant. Data represent mean ± SD; the statistical difference was assessed with *t* test, *P* value as indicated. Twenty-one cross-sections in the midpiece region were analyzed for WT and 21 for *Cfap44*$^{-/-}$. **d**–**f** Various structural defects of the axoneme in sperm from *Cfap44*$^{-/-}$ males. **d** Four DMTs (4–7) were missing. The central pair is shifted at the periphery (red arrowhead). **e** DMTs 5–7 were missing. Note the presence of a third longitudinal column (LC). **f** Irregular distribution of the DMTs associated with a rotation of the central pair (straight red line). Scale bars = 196 nm. **g** In WT sperm, the fibrous sheath is linked to the 3-central-8 complex by stalks emerging from the longitudinal columns (white arrowheads). **h**, **i** The presence of extra ODFs facing DMTs 3 and 8 (red arrowheads) prevents a normal anchoring of the fibrous sheath's stalks on DMTs 3 and 8. Scale bars = 270 nm. **j**–**o** LCs are not aligned with 3–8 CPC axis. In contrast to WT, where the 3-central-8 complex is aligned with LC to form almost a straight line (**j**, red line), LC are misaligned in sperm from *Cfap44*$^{-/-}$ males, leading to notable asymmetry of the structure (**k**, **m**, red lines). **l**, **n** The presence of a third LC increases asymmetry. Scale bars = 184 nm. **o** Bar graph showing the % of defects observed in the principal piece as described in **g**–**n**. One hundred cross-sections in the principal piece region were analyzed for WT and 50 for *Cfap44*$^{-/-}$

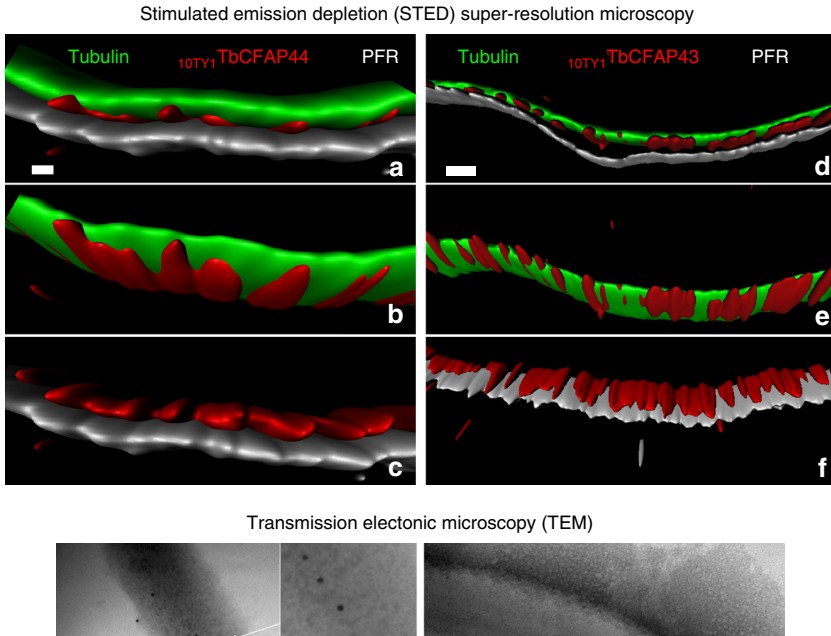

Stimulated emission depletion (STED) super-resolution microscopy

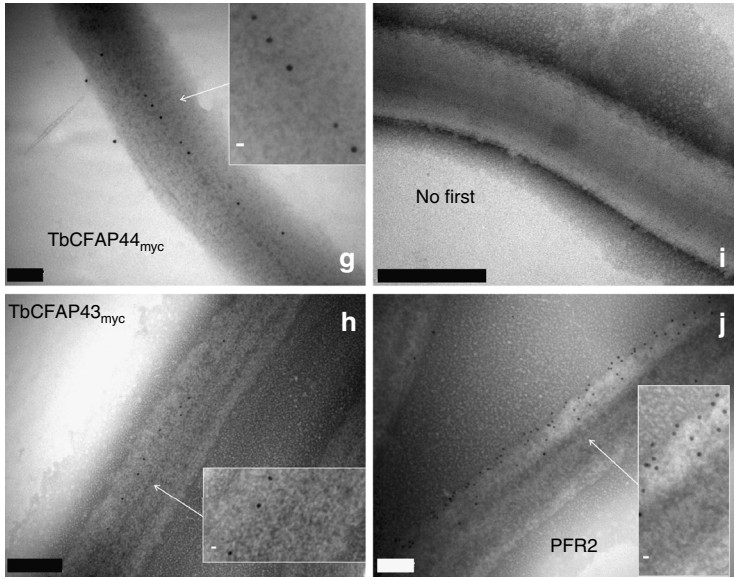

Transmission electonic microscopy (TEM)

**Fig. 7** Orthologs of *T. brucei* TbCFAP44 and TbCFAP43 are flagellar proteins apposed to DMTs 5–6. **a–f** STED confocal microscopy, deconvolution, and 3D reconstruction views of the co-labeling, on permeabilized whole cells, of tubulin (green), PFR2 (gray), and ₁₀ₜᵧ₁TbCFAP44 (**a–c**) and ₁₀ₜᵧ₁TbCFAP43 (**d–f**) showing that TbCFAP44 and TbCFAP43 are facing the PFR and are closely associated with the tubulin, which likely corresponds to DMTs 5–6. Scale bars=0.5 µm. **g–j** Electron immunogold labeling of cytoskeleton-extracted cells expressing TbCFAP44myc (**g**) and TbCFAP43myc (**h**). Controls with no primary antibody (**i**) and anti-PFR2 (**j**). Gold beads size was either 6 nm (**g**, **h**) or 10 nm (**i**, **j**). Scale bars=200 nm in **g**, **h**; 100 nm in **i**, **j**. Insets are enlargement of images of flagella taken from the main panels and scale bars=10 nm

axoneme, composed of nine DMTs and the CPC, is observed (Fig. 8r). In control cell line only a small proportion (0.6%) of axonemes (*n* = 172) were abnormal. In contrast, *TbCFAP44*^RNAi-induced and *TbCFAP43*^RNAi-induced cells were abnormal and displayed more than two flagella in one abnormally enlarged flagellar pocket (Fig. 8p, q). In cross-section, flagella exhibited numerous abnormal axonemes with 90° rotated CPC, a defect never observed in control trypanosome cell lines (Fig. 8p) and mainly displaced CPC and DMTs (Fig. 8s, t). Overall 27.3% of *TbCFAP44*^RNAi (*n* = 139) and 27.7% of the *TbCFAP43*^RNAi (*n* = 150) showed axonemal defects. We could not exclude some minor modifications in PFR; however, the disruption of the axoneme was so extreme after TbCFA44 or TbCFA43 RNAi knockdown that it was difficult to assess them. Altogether, these results confirm the essential role of TbCFAP43 and TbCFAP44 in the organization of the axoneme ultrastructure.

## Discussion

Our work illustrates the efficiency of the combination of WES with an original workflow for the validation of the candidate genes. We identified two new genes responsible for MMAF syndrome and several new *DNAH1* mutations, reinforcing the importance of this gene in MMAF syndrome. The prevalence of *DNAH1* mutations was lower than what was reported previously, maybe due to a wider geographic recruitment of patients in the current study[5,6]. Altogether, *DNAH1*, *CFAP43*, and *CFAP44* mutations were identified in 28.2% of the analyzed subjects (*n* = 78) originating from North Africa, Europe, and the Middle East. These results underline the importance of these three genes in MMAF syndrome and will permit to improve the diagnosis efficiency of infertile MMAF patients. To investigate the possibility of a potential genotype–phenotype correlation, we examined the semen parameters of patients carrying mutations in *DNAH1*, *CFAP43*, and *CFAP44* and MMAF patients with yet

unidentified mutations. There were not differences between the first three groups except for a higher rate of spermatozoa with short and absent flagella and a lower motility rate evidenced in the mutated subjects compared to patients with no identified mutation, highlighting the severity of mutations in these three genes. The identified mutations are distributed throughout the whole CFAP43 and 44 genes indicating that the entirety of both proteins is necessary for preserving their functionality. Interestingly among *CFAP43*-mutated patients, only one (P43-9) carried a missense variant, with potentially a milder effect (Val347Ala). This patient has the lowest percentage of sperm with short flagella (22%) with 10% of morphologically normal spermatozoa (versus 0% for the other *CFAP43*-mutated subjects) and therefore presents a milder phenotype compared to the others with

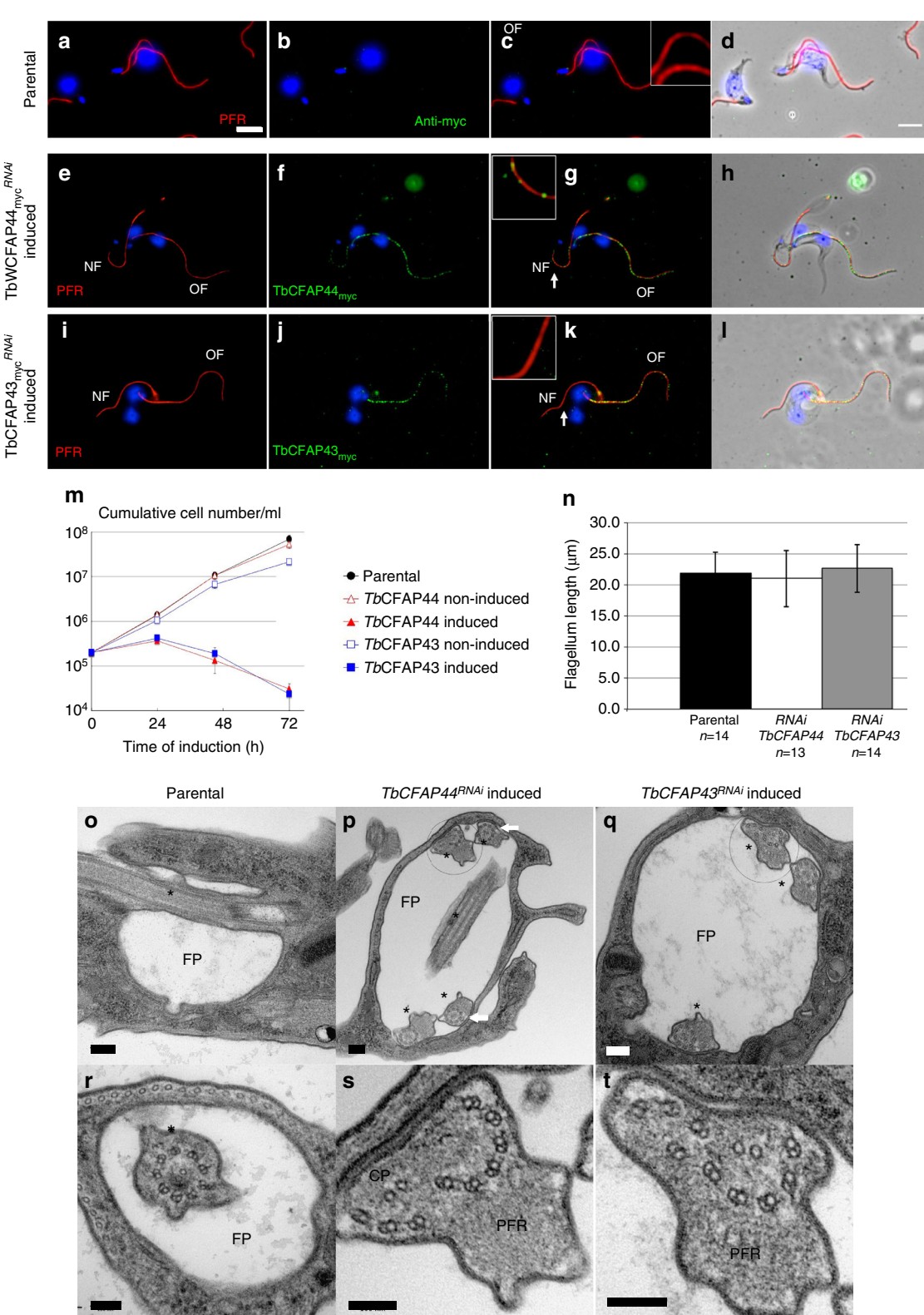

homozygous truncating mutations. These data support the existence of a genotype–phenotype correlation as had been demonstrated for *DNAH1*-mutated subjects[5,6]. We have previously demonstrated that MMAF patients with *DNAH1* mutations had a good prognosis using ICSI[37], but it remains difficult to predict the success rate for the other MMAF genotypes. Additional correlation studies should now be performed to take into account the individual genotypes in the counseling of MMAF patients. Exome sequencing of MMAF patients permitted a diagnosis efficiency of 27% for the three validated genes. Subsequent analysis of exome data from the remaining 56 subjects permitted to identify 7 additional candidate genes with bi-allelic mutations found in at least two individuals in a total of 23 subjects. The identified mutations and these additional genes are currently being investigated. If all these variants are confirmed to be deleterious, a diagnosis will be obtained for 45/78 patients (58%), confirming the interest of WES as a diagnostic tool for MMAF syndrome.

Identification of disease-causing genes has recently been catalyzed by high-throughput genome-wide sequencing technologies. New challenges are now arising, in particular, the validation of candidate genes in the investigated phenotypes. CRISPR/Cas9 is a very efficient and fast technique to create KO models[38], compatible with the increased number of mutations/genes found by high-throughput sequencing. To confirm the pathogenicity of the identified mutations, we produced two knockout mice lines using CRISPR/Cas9 technology. *Cfap43*[−/−] males were infertile and 100% of their spermatozoa were immotile and morphologically abnormal with short and coiled tail, a phenotype very similar to human MMAF (Fig. 4e). *Cfap44*[−/−] male mice were also infertile due to flagellar immotility, yet presenting subtler flagellar defects than those observed in patients with *CFAP44* mutations (Fig. 4f). Such a phenotypic discrepancy between mouse and human is not uncommon. For example, *Dnah1* KO mice display asthenozoospermia without morphological defects of the flagellum[39], whereas in humans *DNAH1* truncating mutations induce a MMAF phenotype[5]. Nevertheless, TEM analysis of *Cfap44*[−/−] sperm evidenced ultrastructural defects similar to those found in *CFAP44* patients, confirming that CFAP44 is necessary for axonemal organization and function both in mouse and human. The reason why this axonemal disorganization does not impact the overall flagellum morphology and length in mice is not known. It is however highly unlikely that a functional truncated protein could persist and have a rescue effect in KO mice since they are predicted at best to produce a truncated peptide of 49 out of the 1,854 amino acids of the full-length protein (Supplementary Fig. 3). Overall, the tandem use of WES and CRISPR/Cas9 technology in mouse

was a very efficient strategy to identify and validate the mutations responsible for infertility phenotypes.

The structure and organization of motile cilia or flagellum was mainly deciphered from models including the green alga *Chlamydomonas reinhardtii*, the protozoa *T. brucei* and *Tetrahymena*, sea urchins, zebrafish, *Xenopus*, and mouse[15,40]. Such a wide selection of distant models is possible because motile cilia or flagella are built on a canonical 9+2 axoneme which forms a highly organized protein network remarkably conserved during evolution[41]. Here we chose to use *T. brucei* as this model has two advantages, first, in contrast to *Chlamydomonas* and other models, it shares some specific structural axonemal characteristics with mammalian flagellum such as a full RS3 and a fixed orientation of the central pair during flagellum beating, and second, it allows forward and reverse genetics for an easier characterization of gene function. BlastP enabled us to identify CFAP43 and CFAP44 likely orthologs in *T. brucei* with good sequence identity. Interestingly, in-depth structural analyses showed that these proteins share some unique structural similarity, in particular, a common general motif with β-strand domains in the N-terminal region followed by α-helical domains in the C-terminal region. These results confirm the relatedness of the identified orthologs and suggest that CFAP43 and CFAP44 proteins may have a similar function as reported for others proteins with similar structures[42]. Consistent with this, we showed that in *T. brucei*, inactivation of one protein by RNAi did not alter the amount and localization of the other protein, indicating that (i) the phenotype observed when invalidating each protein is independent of the other protein, (ii) the presence of one cannot compensate for the absence of the other, and (iii) if the two proteins interact with one another, they do not depend on each other for adequate positioning and docking.

In all studied models (i.e. humans, mouse, and *T. brucei*), the invalidation of each protein induced important axonemal disorganization. It is important to underline that the destabilization of the axoneme can be triggered by the absence of proteins located either in axonemal or peri-axonemal structures[43,44], making it difficult to assess the function from the observed defects. Only the localization at the ultrastructural level would provide a clue on protein function. For this purpose, we generated several *T. brucei* cell lines expressing endogenously tagged proteins, and studied their localization by STED and immuno-EM. Using two different tags, we showed that both proteins have a restricted location within the flagellum, closely associated with the part of the axoneme facing the PFR (Fig. 7) and most probably corresponding to DMTs 5 and 6[35]. This localization rules out a potential role of both proteins in the intraflagellar transport

**Fig. 8** *T. brucei* CFAP44 and CFAP43 are necessary for cell survival and proper axonemal organization. **a–d** Immunofluorescence on detergent-extracted cells of parental *T. brucei* stained with anti-PFR (red) and anti-myc (green) antibodies. **e–h** Immunofluorescence on detergent-extracted cells expressing TbCFAP44myc and RNAi induced (24 h) for *TbCFAP44* (*TbCFAP44myc*[RNAi]). Cells showed no or weak myc labeling (green) on the new flagellum (NF), while the old flagellum (OF) remained labeled. PFR is labeled in red. Note: cells with a maximum of two flagella were chosen for clear imaging. **i–l** Similar RNAi experiments as performed in **e–h** for *TbCFAP43* RNAi in cells expressing TbCFAP43myc (*TbCFAP43myc*[RNAi]). Cells showed no or weak myc labeling (green) on the new flagellum (NF), while the old flagellum (OF) remained labeled. PFR is labeled in red. Nuclei and kinetoplasts (mitochondrial genome) are labeled with DAPI (blue). Scale bars=5 μm. Insets are enlargement of images of flagella taken from the main panels and display areas indicated by white arrows (scale bar=1 μm). **m** Growth curves for parental cells, and *TbCFAP44*[RNAi] and *TbCFAP43*[RNAi] cells, non-induced or induced with tetracycline. The graph represents the cumulative number of cells per ml. Error bars represent the standard error from three independent experiments. **n** Flagellum length measurement of the new flagellum in cells bearing two flagella from parental cells, and from cells expressing TbCFAP44myc and TbCFAP44myc and induced (24 h) for *TbCFAP44* RNAi and *TbCFAP43* RNAi, respectively. Flagellum length was measured in cells showing a clear decrease in myc labeling in the new flagellum. **o–t** Electron micrographs of stained thin sections of parental cells (**o**, **r**) bearing one flagellar pocket (FP) and one flagellum (*), and of 24 h induced *Tb*CFAP44[RNAi] (**p**, **s**) and *Tb*CFAP43[RNAi] (**q**, **t**) cells. **s** and **t** are enlargements of **p** and **q**, respectively. Scale bars=100 nm in **o–q**; 200 nm in **r–t**. Note: in *Tb*CFAP44[RNAi] and *Tb*CFAP43[RNAi] cells, the flagellar pocket is enlarged and bears more than two flagella (**p**, **q**). Some of these flagella present axonemal defects including displaced and rotated CPC and shifted DMTs (**p**, white arrows, **s**, **t**)

(IFT) since IFT in *T. brucei* flagella was never described between the PFR and the axoneme and is circumscribed to two sets of doublet microtubules 3–4 and 7–8, located on each side of the PFR[45]. Moreover, the absence of cytoplasmic material accumulated at the basis of the flagellum and in the flagellar pocket and the normal length of the flagellum in both RNAi mutants (Fig. 8) do not support the IFT hypothesis[46].

Interestingly, it has been reported that two specific structures are present around DMTs 5 and 6: the first one is proposed to consolidate the interaction between the axoneme and the para-axonemal components[47] and the second one to specifically strengthen DMTs 5–6 interaction, known as the 5–6 bridge[11], made of inner and outer subunits. Both structures, the 5–6 bridge and the connecting proteins linking the axoneme to the PFR, are known to present an intervallic pattern[11,47]. This subcellular distribution is in agreement with our super-resolution and immuno-EM results which showed a minimum interval between two immunogold particles of around 25 nm for both TbCFAP44 and TbCFAP43, a distance similar to the interval between two outer 5–6 bridge subunits observed in the flagellum of sea urchin sperm[11]. All these connecting structures, located in the same area, could share physical interactions and the absence of one of the CFAP proteins could destabilize the whole complex, leading to both para-axonemal and axonemal defects. This echoes the anomalies evidenced in IF and TEM experiments in both mouse models and in man. This was particularly noticeable in *Cfap44*[−/−] sperm, where DMTs 5–6 were preferentially missing and para-axonemal structures such as dense fibers and FSs were defective (Fig. 6). Altogether, these results suggest that TbCFAP44 and TbCFAP43 may be the first two proteins specifically located next to DMTs 5–6, and therefore opening the way for a fine characterization of the protein complexes located in this area.

Apart from infertility, the studied patients did not present any obvious PCD-associated symptoms, such as cough, rhinitis, sinusitis, and rhinorrhea and chronic bronchitis. It is not very surprising because flagellar and cilia defects are not always associated[48]. We already have shown that it is also true for *DNAH1*-mutated patients[5,6,49,50]. These observations suggest that mutations in MMAF genes are only responsible for primary infertility without other PCD features and reinforce the presumption that the sperm flagellum is assembled and organized through a specific way, different from other cilia. These differences are notably illustrated by a different beating pattern and the presence of specific peri-axonemal structures, mainly the FS and the ODFs[51]. Importantly, we demonstrated here that in the trypanosome, CFAP43 and CFAP44 are associated to a subset of microtubule doublets facing the PFR. The PFR does not exist in mammalian sperm but its function is likely covered by the FS and the ODF. The fact that the absence of CFAP43/44 has no visible impact on motile cilia could therefore be explained by the functional specificity of the proteins: to interact/link the axoneme with flagella specific extra-axonemal structures, not present in motile cilia in respiratory tissues, brain, or in the oviduct. Further work should now be performed to elucidate the differential composition and organization of motile cilia and flagella. Last, analysis of KO heterozygous mice showed a small deterioration of sperm motility and morphology, suggesting that heterozygous mutations in key spermatogenesis genes might, alone or more likely through cumulative genetic and/or environmental factors, contribute to the less severe but much more frequent phenotype of mild to intermediate oligoasthenozoospermia.

## Methods

**Subjects and controls**. We included 78 subjects presenting with asthenozoospermia due to a combination of morphological defects of the sperm flagella including: absent, short, bent, coiled flagella, and of irregular width without any of the additional symptoms associated with primary ciliary dyskinesia (PCD). The morphology of patients' sperm was assessed with Papanicolaou staining. Small variations in protocol might occur between the different laboratories. Subjects were recruited on the basis of the identification of >5% of at least three of the aforementioned flagellar morphological abnormalities (absent, short, coiled, bent, and irregular flagella).

The global average of all semen parameters are presented in Table 1 and were compared between the different genotype groups using a two-tailed *t* test. Forty-six patients were of North African origin and consulted for primary infertility at the Clinique des Jasmin in Tunis. Ten individuals originated from the Middle East (Iranians) and were treated in Tehran at the Royan Institute (Reproductive Biomedicine Research Center) for primary infertility and 22 patients were recruited in France: 21 at the Cochin Institute and 1 in Lille.

All patients were recruited between 2008 and 2016. All subjects had normal somatic karyotypes. Approximately half of the patients declared to be born from related parents. Sperm analysis was carried out in the source laboratories during the course of the routine biological examination of the patient, according to World Health Organization (WHO) guidelines[52]. Saliva and/or peripheral blood was obtained for all participants. During their medical consultation, all subjects answered a health questionnaire focused on PCD manifestations for infertility. Informed consent was obtained from all the subjects participating in the study according to the local protocols and the principles of the Declaration of Helsinki. In addition, the study was approved by local ethic committees. The samples were then stored in the CRB Germetheque (certification under ISO-9001 and NF-S 96-900) following a standardized procedure. Controls from fertile individuals with normal spermograms were obtained from CRB Germetheque. Consent for CRB storage was approved by the CPCP Sud-Ouest of Toulouse (coordination of the multi-sites CRB Germetheque).

**WES and bioinformatics analysis**. Genomic DNA was isolated from saliva using Oragen DNA Extraction Kit (DNAgenotech®, Ottawa, ON, Canada). Coding regions and intron/exon boundaries were enriched using the all Exon V5 Kit (Agilent Technologies, Wokingham, UK). DNA sequencing was undertaken at the Genoscope, Evry, France, on the HiSeq 2000 from Illumina®. Sequence reads were aligned to the reference genome (hg19) using MAGIC[53]. MAGIC produces quality-adjusted variant and reference read counts on each strand at every covered position in the genome. Duplicate reads and reads that mapped to multiple locations in the genome were excluded from further analysis. Positions whose sequence coverage was below 10 on either the forward or reverse strand were marked as low confidence, and positions whose coverage was below 10 on both strands were excluded. Single-nucleotide variations and small insertions/deletions (indels) were identified and quality-filtered using in-house scripts. Briefly, for each variant, independent calls are made on each strand, and only positions where both calls agree are retained. The most promising candidate variants were identified using an in-house bioinformatics pipeline, as follows. Variants with a minor allele frequency >5% in the NHLBI ESP6500 (Exome Variant Server, NHLBI GO Exome Sequencing Project (ESP), Seattle, WA) or in 1,000 Genomes Project phase 1 datasets[54], or >1% in ExAC[55], were discarded. We also compared these variants to an in-house database of 94 control exomes obtained from subjects mainly originated from North Africa and Middle East corresponding to the geographical origin of most patients from this study and which is under-represented in SNP public databases. All variants present in homozygous state in this database were excluded. We used Variant Effect Predictor (VEP version 81[56]) to predict the impact of the selected variants. We only retained variants impacting splice donor/acceptor or causing frameshift, inframe insertions/deletions, stop-gain, stop loss, or missense variants, except those scored as "tolerated" by SIFT[57] (sift.jcvi.org) and as "benign" by Polyphen-2[58] (genetics.bwh.harvard.edu/pph2). Finally, identified mutations were validated by Sanger sequencing. PCR primers and protocols used for each patient are listed in the Supplementary Table 3. Sequencing reactions were carried out with BigDye Terminator v3.1 (Applied Biosystems). Sequences analysis were carried out on ABI 3130XL (Applied Biosystems). Sequences were analyzed using SeqScape software (Applied Biosystems).

**RT-qPCR analysis**. RT-qPCR was performed with cDNAs from various tissues of human and mouse including testes. A panel of eight organs was used for mouse experiments: testis, brain, lung, kidney, liver, stomach, colon, and heart. RNA extraction were performed from three DBA-C57 WT mice with the mirVana™ PARIS™ Kit (Life Technologies®). For human experiments, a panel of the three main ciliated tissues was used: testis, brain, and lung. Human RNAs were purchased from Life Technologies®. Reverse transcriptions were performed using the High Capacity cDNA Reverse Transcription Kit (Applied Biosystem®). Each sample was assayed in triplicate for each gene on a StepOnePlus (Life Technologies®), with Power SYBR®Green PCR Master Mix (Life Technologies®). The PCR cycle was as follows: 10 min at 95 °C, 1 cycle for enzyme activation; 15 s at 95 °C, 60 s at 60 °C with fluorescence acquisition, 40 cycles for the PCR. RT-qPCR data were normalized using the reference housekeeping gene *ACTB* for human and mouse with the $-\Delta\Delta Ct$ method[59]. The $2-\Delta\Delta Ct$ value was set at 0 in brain cells, resulting in an arbitrary expression of 1. Primers sequences and RT-qPCR conditions are indicated in the Supplementary Table 4. The efficacy of primers was checked using a standard curve. Melting curve analysis was used to confirm the presence of a single PCR product. Statistics were performed using a two-tailed *t* test on the Prism 4.0 software (GraphPad, San Diego, CA, USA) to compare the relative expression of *CFAP43* and *CFAP44* transcripts in several organs. Statistical tests with a two-tailed *P* values ≤0.05 were considered significant.

**Immunostaining in human and mouse sperm cells**. We performed IF staining on human and mouse spermatozoa as described by our laboratory[60]. In human, immunostaining could be carried out on samples from two of the patients with a stop-gain mutation, one mutated in each gene. IF experiments were performed using sperm cells from control individuals, from the patient $P_{43}$-5 homozygous for the c.2658C>T variant in CFAP43 and from the patient $P_{44}$-2 homozygous for the c.3175C>T variant in CFAP44. Sperm cells were fixed in phosphate-buffered saline (PBS)/4% paraformaldehyde for 1 min at room temperature. After washing in 1 ml PBS, the sperm suspension was spotted onto 0.1% poly-L-lysine pre-coated slides (Thermo Scientific). After attachment, sperms were permeabilized with 0.1% (v/v) Triton X-100–DPBS (Triton X-100; Sigma-Aldrich) for 5 min at room temperature. Slides were then blocked in 5% corresponding normal serum–DPBS (normal goat or donkey serum; Gibco, Invitrogen) and incubated overnight at 4 °C with primary antibodies. For human experiments, the following primary antibodies were used: DNAI2, DNALI1, RSPH1, RPSH4A, SPAG6, GAS8, AKAP4, and anti-acetylated-α-tubulin. For mouse experiments: Rsph1, Rsph4a, Gas8, Spef2, Dnali1, Dnah5, Mpc1l, and anti-acetylated-α-tubulin. Primary antibodies references, provider, species, and dilutions used are listed in the Supplementary Table 5. Washes were performed with 0.1% (v/v) Tween-20–DPBS, followed by 1 h incubation at room temperature with secondary antibodies. Highly cross-adsorbed secondary antibodies (Dylight 488 and Dylight 549) were from Jackson Immunoresearch®. Appropriate controls were performed, omitting the primary antibodies. Samples were counterstained with 5 mg/ml Hoechst 33342 (Sigma-Aldrich) and mounted with DAKO mounting media (Life Technology). Fluorescence images were captured with a confocal microscope (Zeiss LSM 710).

**Electron microscopy of human and mouse sperm cells**. We performed TEM experiments on human and mouse ($Cfap43^{-/-}$ and $Cfap44^{-/-}$ and WT) spermatozoa. In human, TEM experiments were performed using sperm cells from fertile control individuals, from patient $P_{43}$-8 homozygous for the c.3352C>T variant in CFAP43 and from patient $P_{44}$-3 homozygous for the c.2818dupG variant in CFAP44. Sperm cells were fixed with 2.5% glutaraldehyde in 0.1 M sodium phosphate (pH 7.4) during 2 h at room temperature. Cells were washed with buffer and post-fixed with 1% osmium tetroxide in the same buffer for 1 h at 4 °C. After washing with distilled water, cells were stained overnight at 4 °C with 0.5% uranyl acetate (pH 4.0). Cells were dehydrated through graded alcohol (30, 60, 90, 100, 100, and 100; 10 min for each bath) and infiltrated with a mix of 1:1 Epon/alcohol 100% for 2 h before two baths of fresh Epon for 2 h. Finally, cells were included in fresh Epon and polymerized during 2 days at 60 °C. Ultrathin sections (90 nm for human samples and 60 nm for mouse samples) of the cell pellet were done with an ultramicrotome (Leica). Sections were post-stained 10 min with 5% uranyl acetate (pH 4.0), washed four times with distilled water (1 h), and then stained with 0.4% lead citrate before being observed in a TEM at 80 kV (JEOL 1200EX), and images were acquired with a digital camera (Veleta, Olympus).

**Trypanosoma cultures and transfection**. The trypanosome cell lines used in this study derived from the bloodstream form parental T. brucei 427 90-13 strain co-expressing the T7 RNA polymerase and tetracycline repressor[61]. Cells were cultured at 37 °C and 5% $CO_2$ and transfected and cloned as described in ref. [62] in medium supplemented with puromycin (0.1 μg/ml) or blasticidin (10 μg/ml) for constitutive expression of myc-tagged proteins and 10TY1-tagged proteins respectively, and with phleomycin (2.5 μg/ml) for RNAi-inducible cells. RNAi interference was induced with tetracycline (10 μg/ml).

**Cell lines generated for this study**. For WT level of expression of 3 x myc C-terminal tagged proteins ($TbCFAP44_{myc}$, $TbCFAP43_{myc}$), parental cells were transfected as in ref. [62] with a tagging cassette that was obtained by PCR using a set of long primers containing 80 nucleotides from the 5′-UTR and 3′-UTR flanking regions of the TbCFAP44 and TbCFAP43 open reading frame (ORF) and pMO-Tag23M vector as template[20] (Supplementary Table 6). For RNAi, fragments of TbCFAP44 ORF (bp 1,880–2,309) and TbCFAP43 ORF (bp 131–596) were cloned into p2T7tiB[63] and transfected into $TbCFAP44_{myc}$, $TbCFAP43_{myc}$ cell lines generating the cell lines $TbCFAP44_{myc}$RNAi and $TbCFAP43_{myc}$RNAi (Supplementary Table 6). For WT level of expression of 10 x TY1 N-terminal-tagged proteins, cell lines expressing $TbCFAP44_{myc}$RNAi and $TbCFAP43_{myc}$RNAi were transfected with a tagging cassette that was obtained by PCR using a set of primers amplifying the end of the 5′-UTR and the beginning of, respectively, the TbCFAP43 and TbCFAP44 ORF and the pPOTv4-10TY1-blast vector as template[34] (Supplementary Table 6).

**Trypanosoma IF**. Cells were collected, washed, and processed for immunolabeling on cytoskeletons (from detergent-extracted cytoskeleton) as in ref. [33], except that the cytoskeletons were fixed in methanol at −20 °C for 30 min. Samples were incubated with the primary antibodies anti-myc and rabbit anti-PFR2 (PFR2 is a protein of the para-axonemal structure called PFR, 1:2,000 dilution in PBS) and with secondary antibodies anti-mouse FITC conjugated (Sigma F-2012, 1:400 dilution in PBS) and anti-rabbit Alexa-594 conjugated (Thermo Fisher A11012, 1:400 dilution in PBS). Nuclei and kinetoplasts were stained with DAPI (4′,6-diamidino-2-phenylindole; 10 μg/ml). Images were acquired on a Zeiss Imager Z1 microscope, using a Photometrics

Coolsnap HQ2 camera, with Zeiss ×100 or ×63 objectives (NA 1.4) using the Metamorph software (Molecular Devices), and processed with ImageJ.

For STED analysis on whole cells, cells were collected, washed, and loaded on poly-L-lysine-coated coverslips and air-dried. After a 5 min rehydration step in PBS, cells were fixed in methanol at −20 °C for at least 30 min. Samples were incubated with antibodies in four steps: (1) with the primary antibody anti-TY1 (BB2)[32], a mouse IgG1 monoclonal, (2) followed by the secondary antibody Alexa-594 coupled anti-mouse IgG1, (3) anti-tubulin (TAT1)[64], a mouse IgG2a monoclonal together with anti-PFR2 (rabbit polyclonal), (4) Alexa-488-coupled anti-mouse IgG2a and ATTO647N-coupled anti-rabbit secondary antibodies. Coverslips were mounted using ProLong Gold antifade reagent. Images were acquired on a Leica DMI6000 TCS SP8 X-STED microscope with a ×93 glycerol objective (NA 1.3), de-convolved with Huygens Pro 16.10, and 3D reconstructions were generated using Imaris X64 8.1.2. Primary antibodies' references, provider, species, and dilutions used are listed in Supplementary Table 6.

**Electron microscopy of Trypanosoma cells**. Cells were fixed in culture medium, by the addition of glutaraldehyde to a final concentration of 2.5%, for 60 min. They were pelleted (1,000 g, 10 min), resuspended in fixation buffer (2.5% glutaraldehyde, 2% PFA, 100 mM phosphate buffer, pH 7.4, 50 mM sucrose for 2 h). Fixed cells were washed in water for 10 min, post-fixed in 1% $OsO_4$ for 1 h, washed three times in water, and then samples were stained in 2% uranyl acetate in water at 4 °C overnight. Samples were next washed in water, dehydrated in ethanol, embedded in Spurr resin, and polymerized overnight at 60 °C. Sections (60–80 nm) were stained in 2% uranyl, then lead citrate, and visualized on a TECNAI 12 TEM.

**Immunogold labeling on detergent-extracted cytoskeleton from Trypanosoma cells**. Cell cultures were pelleted (1,000 g, 10 min), washed in vPBS (PBS, sucrose 15.7 g/l, glucose 1.8 g/l) and extracted with PIPES 100 mM, NP-40 (Igepal) 1%, $MgCl_2$ 1 mM, benzonase. Cytoskeletons were pelleted by centrifugation 10 min at 2,500 g and washed twice in PIPES 100 mM, 1%, $MgCl_2$ 1 mM, benzonase. Two hundred and fifty microliters of droplets of cytoskeletons were placed on parafilm. Butvar covered, charged, carbon-coated nickel grids were floated on each drop and incubated for 10 min to let the cytoskeletons adhere. The grids were then moved through two blocking droplets for 5 min each (blocking buffer—1% Fish skin gelatin, 0.01% Tween-20 in PBS) and then to droplets containing the primary antibody in blocking buffer (anti-TY1 1:10, anti-myc, anti-PFR2 mouse monoclonal L8C4) for 2 h at room temperature. The grids were washed four times in blocking buffer and incubated with gold-conjugated secondary antibodies—EM GAR10 1:20 (anti-mouse, 6 nm), in blocking buffer for 2 h at room temperature. After incubation, grids were washed 3× for 5 min in blocking buffer, 2× 5 min in 0.1% fish skin gelatin, 0.001% Tween-20 in PBS, 2× for 5 min in PBS, and then fixed in 2.5% glutaraldehyde in milliQ $H_2O$ for 5 min. Samples were negatively stained with 10 μl auro-thioglucose for 20 s. Samples were visualized on a FEI Tecnai 12 electron microscope, camera ORIUS 1000 11M Pixel (resolution 3–5 nm). Images were acquired with Digitalmicrograph and processed with ImageJ.

**Western blotting on detergent-extracted cytoskeleton from Trypanosoma cells**. Proteins from detergent-extracted cytoskeleton ($2.10^7$) were separated on sodium dodecyl sulfate-polyacrylamide gel electrophoresis gels (6%) and semi-dry transferred (Bio-Rad) for 45 min at 25 V on PVDF membrane. After a 1 h blocking step in 5% milk in PBS-0.2% Tween-20, the membranes were incubated overnight at 4 °C with the anti-TY1 primary antibody (BB2, 1 :5,000) diluted in blocking buffer. After three washes in blocking buffer and one wash in 1 M NaCl, the membranes were incubated with the HRP-conjugated secondary antibody (Jackson, 1:10,000 dilution), washed twice 10 min in blocking buffer and twice 5 min in PBS. Blots were revealed using the Clarity Western ECL Substrate Kit (Bio-Rad) with the ImageQuant LAS4000. After membrane stripping, protein loading was controlled by probing tubulin (TAT1, 1:500) as described above.

**CRISPR/Cas9 KO mice**. All animal procedures were run according to the Swiss and French guidelines on the use of animals in scientific investigations with the approval of the French local Ethical Committee (ComEth Grenoble No. 318, ministry agreement number #7128 UHTA-U1209-CA) and the Direction Générale de la Santé (DGS) for the State of Geneva. All the procedures were done in Geneva until the birth of the modified litters. For each gene, three plasmids were injected directly into the nucleus of the zygotes. One plasmid expressed the Cas9 protein and the other two expressed two distinct RNA guides (single guide RNA, or sgRNA) targeting exons 2 and 21 of the Cfap43 gene and exons 3 and 15 of the Cfap44 gene. All plasmids (pGS-U6-sgRNA expression vector) were ordered from GeneScript (Piscataway, NJ, USA) with the different cDNA sequences already inserted. sgRNA sequences are indicated in the Supplementary Table 7. The Cas9 nuclease and sgRNAs were introduced into fertilized oocytes. Microinjected oocytes were introduced into pseudopregnant host females and carried to term. Edited founders were identified by Sanger sequencing from tail biopsies. Tail biopsies (2 mm in length) were digested in 200 μl lysis Direct PCR Lysis Reagent (Tail) (Viagen Biotech Inc., Los Angeles, CA, USA) and 0.2 mg of proteinase K for 12–15 h at 55 °C and 1 h at 85 °C. The DNA was directly used for PCRs. The two targeted exons were amplified using the following PCR protocol: 59 °C x 1 cycle,

58 °C x 1 cycle, and 57 °C x 35 cycles with 1 min elongation. Sanger sequencing was then performed to identify CRISP/Cas9-induced mutations and genotypes were determined according to the sequence electrophoregrams. Mice carrying desired modification events are bred with C57BL6/J to ensure germline transmission and eliminate any possible mosaicism. From the second generation of mice, genotyping was performed by high-resolution melting (HRM) using MeltDoctor™ HRM Master Mix with the following parameters: 10 min at 95 °C for enzyme activation, 15 s at 95 °C for denaturation, 1 min at 60 °C for annealing and extension, then for the melting curve, 1 min at 95 °C for denaturation, 1 min at 60 °C for annealing, 15 s at 95 °C for HRM and 15 s at 60 °C for annealing. List of primers used for mice genotyping with both methods is available in the Supplementary Table 8.

Heterozygous animals were mated to generate homozygous offspring. Approximately 25% of the offspring were homozygous for the mutated allele, indicating the absence of an increased embryonic or postnatal lethality in $Cfap43^{-/-}$ and $Cfap44^{-/-}$ mice. Mice were housed with unlimited access to food and water and were sacrificed by cervical dislocation after 8 weeks old, which means that they were pubescent and that their reproductive organs were fully established.

The edited gene expression in mutant mice was validated by RT-PCR followed by Sanger sequencing of testicular transcripts. RT-PCR experiments were performed using testis RNA from WT, heterozygous, and homozygous animals. Reverse transcription was carried out with 5 μl of extracted RNA (~500 ng) using Macherey Nagel columns (Macherey Nagel, Hoerdt, France) according to the manufacturer's protocol. Hybridization of the oligo(dT) was performed by incubating for 5 min at 65 °C and quenching on ice with the following mix: 5 μl RNA, 3 μl of poly T oligo primers (dT) 12–18 (10 mM, Pharmacia), 3 μl of the four dNTPs (0.5 mM, Roche Diagnostics), and 2.2 μl of $H_2O$. Reverse transcription then was carried out for 30 min at 55° after the addition of 4 μl of 5× buffer, 0.5 μl RNase inhibitor, and 0.5 μl of Transcriptor reverse transcriptase (Roche Diagnostics). Two microliters of the obtained cDNA mix was used for the subsequent PCR and Sanger sequencing. Primers sequences and RT-PCR conditions are indicated in the Supplementary Table 9.

**Phenotypic analysis of mouse mutant**. To test the fertility, pubescent $Cfap43^{-/-}$ and $Cfap44^{-/-}$ males (8 weeks old) were mated with C57BL6/J females. To determine sperm concentration, sperm samples were collected from the cauda epididymis of 8-week-old $Cfap43^{-/-}$ and $Cfap44^{-/-}$ and sperm number was determined using a hemocytometer under a light microscope. For sperm morphology, sperm was displayed over a slide, dried at room temperature, and then fixed in 75% ethanol. Harris–Schorr staining was performed according to the WHO protocol. Schorr staining solution was obtained from Merck and at least 100 sperm per animal were analyzed. Mobility of sperm was assessed with computer-assisted motility analysis (CEROS I, Hamilton Thorn Research, Beverly, MA, USA) in an analysis chamber (100 μm depth, 30 μl volume, Leja Products B.V., Netherlands) at 37 °C. The settings employed for the analysis were as follows: acquisition rate, 60 Hz; number of frames, 100; minimum contrast, 25; minimum cell size, 10; low static size gate, 2.4; high static size gate, 2.4; low static intensity gate, 1.02; high static intensity gate, 1.37; minimum elongation gate, 12; maximum elongation gate, 100; magnification factor, 0.70. The motility parameters measured were curvilinear velocity, average path velocity (VAP), and straight line velocity (VSL). At least 100 motile sperm were analyzed for each assay. Motile sperm and progressive sperm were characterized by VAP >1 μm/s, by average path velocity >30 μm/s and straightness (VSL/VAP) >70%, respectively. For each genotype, five mice were used.

Statistical analyses were performed with the SigmaPlot software. Unpaired $t$ test was used to compare the different genotypes. Statistical tests with a two-tailed $P$ values ≤0.05 were considered significant.

**RNA expression analysis**. Total RNA was isolated from $10^8$ cells of parental, non-induced, and tetracycline-induced RNAi cells using the TRIzol reagent according to the manufacturer's instructions (Life Technologies). The constitutively expressed housekeeper telomerase reverse transcriptase was used as an internal control[65]. RT-PCR was carried out with a SuperScript III One-step RT-PCR System with Platinum $Taq$ high-fidelity polymerase (Life Technologies Ltd, Paisley, UK) following the manufacturer's protocol. Briefly, 100 ng of total RNA were mixed with primers and reverse transcriptase solution in 50 μl total volume and using the following cycle protocol: 30 min at 55 °C (reverse transcription); 2 min at 94 °C (inactivation of reverse transcriptase and activation $Taq$ polymerase); followed by 25 cycles of regular PCR (denaturation: 94 °C for 15 s; annealing: 55 °C for 30 s; extension: 68 °C for 40 s); and finalized with a hold for 5 min at 68 °C. The RT-PCR products were resolved on 1% agarose running gel with BET and visualized by UV light. Primer sequence is provided in Supplementary Table 10.

**Data availability**. Data on genetic variants described here are available on ClinVar (https://www-ncbi-nlm-nih-gov.gate2.inist.fr/clinvar/) – submission ID SUB2319254444. All other data that support the findings of this study are available from the corresponding author upon reasonable request.

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

## Acknowledgements

We thank Anne Bertrand from the GIN electron microscopy platform and the IAB microscopy platform and Alexei Grichine and Jacques Mazzega for their technical help. We thank Klaus Ersfeld (Bayreuth University) for the anti-myc antibody, Nicolas Biteau (Bordeaux University) for the anti-PFR2 antibody, Keith Gull (Oxford University) for the TAT1 antibody, Philippe Bastin (Institut Pasteur, Paris) for the anti-TY1 antibody, and Vanderperre Benoît and Martinou Jean Claude (UNIGE university, Geneva, CH) for the Mpcl1l antibody. We thank Denise Escalier and Gérard Gascon for discussions and expertise on the topic. The STED microscopy was done in the Bordeaux Imaging Centre a service unit of the CNRS-INSERM and Bordeaux University, member of the national infrastructure France BioImaging supported by the French National Research Agency (ANR-10-INBS-04). The help of Christel Poujol and Patrice Mascalchi is acknowledged. Some of the electron microscopy was done at the Bordeaux Imaging Centre. We thank our patients and control individuals for their participation. This work was mainly supported by the following grants: The "MAS-Flagella" project financed by French ANR and the DGOS for the program PRTS 2014 and the "Whole genome sequencing of patients with Flagellar Growth Defects (FGD)" financed by the fondation maladies rares (FMR) for the program Séquençage à haut débit 2012.

## Author contributions

C.C., M.B., A.T., C.A., and P.F.R. analyzed the data and wrote the manuscript; Z.-E.K., A.A.-Y., L.S., P.L., E.E.K., J.F., and C.C. performed molecular work; T.K., J.-F.D., A.B., and N.T.-M. analyzed genetic data; C.C., A.S.V., P.L.T., C.W.-L., and S.P.B. performed IF experiments; S.C. performed protein in silico analysis, K.P.-G., A.S.V, A.S., and A.T. performed EM experiments, Z.-E.K., A.A.-Y., A.S.V., B.C., S.F.B.M., S.N., and J.E. performed mouse work, D.D., N.L., D.R.R., and M.B. performed *Trypanosoma* work. N.L. and M.B. performed STED analysis. S.F.B.M., G.M., A.D., S.H.H., V.M., L.H., O.M., M.M., H.L., M.K., S.H., V.S., P.-S.J., J.-P.W., H.G., E.D., and R.Z. provided clinical samples and data; M.B., A.T., C.A., and P.F.R. designed the study, supervised all molecular laboratory work, had full access to all of the data in the study. and takes responsibility for the integrity of the data and its accuracy. All authors contributed to the report.

## Additional information

Charles Coutton[1,2], Alexandra S. Vargas [1], Amir Amiri-Yekta [1,3,4], Zine-Eddine Kherraf[1,3], Selima Fourati Ben Mustapha[5], Pauline Le Tanno[1,2], Clémentine Wambergue-Legrand[1,3], Thomas Karaouzène[1,3], Guillaume Martinez[1,2], Serge Crouzy[6], Abbas Daneshipour[4], Seyedeh Hanieh Hosseini[7], Valérie Mitchell[8], Lazhar Halouani[5], Ouafi Marrakchi[5], Mounir Makni[5], Habib Latrous[5], Mahmoud Kharouf[5], Jean-François Deleuze[9], Anne Boland[9], Sylviane Hennebicq[1,10], Véronique Satre[1,2], Pierre-Simon Jouk[11], Nicolas Thierry-Mieg[12], Beatrice Conne[13], Denis Dacheux[14,15], Nicolas Landrein[14], Alain Schmitt[16,17,18], Laurence Stouvenel[16,17,18], Patrick Lorès[16,17,18], Elma El Khouri[16,17,18], Serge P. Bottari[1], Julien Fauré[19,20], Jean-Philippe Wolf[18,21], Karin Pernet-Gallay[20], Jessica Escoffier[1], Hamid Gourabi[4], Derrick R. Robinson[14], Serge Nef[13], Emmanuel Dulioust[18,21], Raoudha Zouari[5], Mélanie Bonhivers [14], Aminata Touré[16,17,18], Christophe Arnoult[1] & Pierre F. Ray [1,3]

[1]Genetic Epigenetic and Therapies of Infertility, Institute for Advanced Biosciences, Inserm U1209, CNRS UMR 5309, Université Grenoble Alpes, 38000 Grenoble, France. [2]CHU de Grenoble, UM de Génétique Chromosomique, 38000 Grenoble, France. [3]CHU de Grenoble, UM GI-DPI, 38000 Grenoble, France. [4]Department of Genetics, Reproductive Biomedicine Research Center, Royan Institute for Reproductive Biomedicine, ACER, Tehran, 16635-148, Iran. [5]Polyclinique les Jasmins, Centre d'Aide Médicale à la Procréation, Centre Urbain Nord, 1003 Tunis, Tunisia. [6]Laboratoire de Chimie et Biologie des Métaux, Institut de Recherche en Technologie et Sciences pour le Vivant, CEA iRTSV/LCBM/GMCT, CNRS UMR 5249, Université Grenoble Alpes, 38054 Grenoble, France. [7]Department of Andrology, Reproductive Biomedicine Research Center, Royan Institute for Reproductive Biomedicine, ACECR, Tehran, 16635-148, Iran. [8]EA 4308, Department of Reproductive Biology and Spermiology-CECOS Lille, University Medical Center, Lille, 59037, France. [9]Centre National de Génotypage, Institut de Génomique, CEA, 91000Evry, France. [10]CHU de Grenoble, UF de Biologie de la procréation, 38000 Grenoble, France. [11]CHU de Grenoble, UF de Génétique Médicale, 38000 Grenoble, France. [12]Univ. Grenoble Alpes/CNRS TIMC-IMAG, 38000 Grenoble, France. [13]Department of Genetic Medicine and Development, University of Geneva Medical School, 1211 Geneva, Switzerland. [14]Microbiologie Fondamentale et Pathogénicité CNRS UMR 5234, University Bordeaux, 33000 Bordeaux, France. [15]Bordeaux-INP, Microbiologie Fondamentale et Pathogénicité, UMR-CNRS 5234, 33000 Bordeaux, France. [16]Institut National de la Santé et de la Recherche Médicale, INSERM U1016, Institut Cochin, 75014 Paris, France. [17]Centre National de la Recherche Scientifique, CNRS UMR8104, 75014 Paris, France. [18]Sorbonne Paris Cité, Faculté de Médecine, Université Paris Descartes, 75014 Paris, France. [19]CHU de Grenoble UF de Biochimie Génétique et Moléculaire, 38000 Grenoble, France. [20]Grenoble Neuroscience Institute, INSERM 1216, 38000 Grenoble, France. [21]Laboratoire d'Histologie Embryologie - Biologie de la Reproduction, GH Cochin Broca Hôtel Dieu, Assistance Publique-Hôpitaux de Paris, 75014 Paris, France. Mélanie Bonhivers, Aminata Touré, Christophe Arnoult, and Pierre F. Ray contributed equally to this work.

