## [Peer Review File · Nature Communications]

Reviewers' comments:

Reviewer #1 (Remarks to the Author):

Re: "Whole exome cohort study and analysis of mouse and Trypanosoma models demonstrate the importance of WDR proteins in flagellagenesis and male fertility" by Coutton et al.

The authors recruited patients showing multiple morphological abnormalities of the flagella (MMAF), which is a severe form of asthenozoospermia. 78 patients were subjected to exome analyses and 16 patients showed mutations in CFAP43 or CFAP44. Most of them are homozygous mutations and several compound heterozygous mutations. Even though each mutation frequency is extremely low, homozygous state of the patients was observed indicating the consanguineous marriage of the parents. The loss of CFAP43 and CFAP44 in human was examined by model system using mouse and Trypanosoma, and very similar phenotypes (indeed it is very surprisingly) were observed.

The authors performed functional analyses of the two genes at intensive and precise manners and the results were well described. Overall I do not have any major comment. I just point out several minor issues.

- 1) In the title, gene name should be shown instead of WDR proteins
- 2) Gene symbol was inconsistently used, for example; DNAH1 (in italic) gene, CFSP43 (non-italic).
- 3) Page 5 lines 140-143; The authors mentioned about variants as a result of negative selection. Negative or positive selection is discussed by gene-based variants (multiple variants in a gene). Indeed heterozygous state seems to be almost normal therefore it seems to be overstatement.
- 4) Page 8 line 218; please add Fig. 4a1-3.
- 5) Page 18, line 471; "determine" should be "determined"
- 6) Page 19 lines 482-484: duplicated

Reviewer #2 (Remarks to the Author):

Review

This interesting manuscript reports the discovery of two novel genes whose mutations are responsible for a male fertility disease called asthenozoospermia. Based on the analysis of patient spermatozoa, of mouse strains and of the model protist Trypanosoma brucei, the authors show that both genes are required for correct architecture of the flagellar apparatus. The paper relies on solid genetic and immunofluorescence analysis, as well as on nice electron microscopy data. The results obtained from mouse and trypanosome studies are quite spectacular. In summary, this work is novel and will be of interest for a large audience (geneticists, cell biologists, people interested in cilia and flagella etc). However, I

believe that the paper can be much improved by providing more quantitative data and by a deeper analysis of the results (see below).

Major points

1. It is intriguing that phenotypes have only been observed in spermatozoa and not in motile cilia of respiratory epithelium. I agree that the expression level is much lower, but the conserved function in trypanosomes would suggest an involvement in all kinds of motile cilia. This point should be discussed.

2. A key question is unsolved in the manuscript: are CFAP43 and CFAP44 required for construction or for maintenance? The trypanosome system is famous for comparing mature and growing flagella. This analysis should be done. It is probably too challenging to do in mice.

3. The discussion is too long (7 pages) and the reader gets lost/confused in multiple arguments and counter-arguments. It needs serious focusing. For example, a contribution of CFAP43 and CFAP44 to the architecture of the central pair is much debated without doing the right experiment that consists in showing association to the central pair by immunogold or biochemical fractionation. I agree it is a difficult experiment, especially without a good antibody, so I can understand if the authors decide not to do it. However, the discussion needs to be refocused to take this into account.

4. TEM images are lovely but numbers are often vague or missing. Table S1 gives % but does not mention how many sections were looked at. I could not find any indications on the number of sections used or the proportion of the various phenotypes for Figures 5, 7 and 8. These numbers must be reported to be able to appreciate the significance of these nice images. I could not find the thickness of the sections either (important parameter knowing the repetitive nature of axoneme associated structures).

5. The trypanosome data are nice but appear underused to me (also see above). Is there a motility phenotype? The authors should show a video as presented for sperm. For the IFA, phase or DIC pictures should be shown to help the reader. Staining patterns for TbCFAP43 and 44 look spotty. Is it related to intraflagellar transport? Does the staining extend from base to tip in mature and growing flagella (not always clear on my images at least)? How related are CFAP43 and 44? The fate of one in RNAi conditions for the other would be interesting to know (i.e. TbCFAP44 in TbCFAP43RNAi and conversely).

6. There are quite a lot of small mistakes in nomenclature or presentation of the data. It is a bit irritating. I made a list of these below but I might have missed some and I therefore strongly advise the authors to check their manuscript carefully.

Other points

Abstract. Nowadays many authors consider cilia and flagella to be different names for equivalent structures. The authors should clearly specify if they talk about the spermatozoa flagellum or other type of cilia and flagella. The way the abstract is written shows some confusion between ciliopathies at large and PCD.

90. Mention the "various geographical origins" of the patients.

92. "indicate"

96. "this cohort" instead of "the cases"

124. Nomenclature: Usually genes are written in italics and proteins in standard. Please check carefully throughout the manuscript.

174. Which prediction softwares were used?

187. It would be more correct to talk about the "skeleton" of cilia and flagella (no real "cyto" here)

189. "complex"

193. "a helical"

236. "reverse" genetics

238. Give a reference for the trypanosome model.

244. What is CMF7?

247. "trypanosome" is a common name (no capital T) but "Trypanosoma" is a Latin word and should be italicised. Please check carefully throughout the manuscript.

251. There are two stages of trypanosomes for these experiments. Which one was used here?

276. The "term knock-out" does not seem appropriate as the gene is still there. Maybe "mutant" would be better.

283. On which criteria these two mouse strains were selected instead of the others? Was it a random selection (if strains really identical) or were there some practical reasons or else? In any case, it should be explained.

290. n should be provided (same lines 296 and 339)

361. zebrafish and Xenopus are two other classic models for the study of motile cilia.

398. CFAP43 and "CFAP44"

480-484. Delete the duplicated sentence.

524. "primary"

848-849. Not clear what is meant by "the green staining of the entire flagellum showing that TbCFAP44 localisation is restricted to the flagellum"? The M&M mentions the use of cytoskeletons (672) so I am not sure if this statement is valid. If there were a soluble pool of TbCFAP43 or 44, it would likely be lost upon detergent extraction.

877. remove "staining"

Table 1. Patients mutated for are indicated with a "+", this is very confusing! Stars mention significance but it is not clear which numbers were compared (no data provided for healthy patients).

Reviewer #3 (Remarks to the Author):

In this paper, the authors have examined 78 patients showing male infertility with morphological abnormalities of sperm flagella (MMAF), by whole-exome sequencing. They identified mutations in DNAH1 (as previously reported), but in addition, they newly identified mutations in CFAP43 (10 patients) and in CFAP44 (6 patients). Analysis of mutant sperm by TEM confirmed MMAF phenotype: structural anomalies such as the absence or mis-orientation of the central pair of microtubules, abnormal organization of peripheral microtubules, disorganized mitochondrial sheath and fibrous sheath etc. They also studied CFAP43 and CFAP44 mutants in Trypanosoma and in the mouse, and found that these mutants show similar defects. In all, this is a very solid paper reporting two evolutionarily conserved genes essential for flagella formation. However, this paper is more suited to a medical genetics journal such as Am. J. Hum. Genet., because no information is available for the function of CFAP43 and CFPA44.

Additional comments (major)

1) It was speculated but CFPA43 and CFAP44 may link central pair microtubules with radial spokes, but precise localization of these proteins in the human or mouse sperm remain unknown. Are they axonemal proteins in the human or mouse? If so, where are they localized in the axoneme? Myc-tagged CFAP43 and CFAP44 proteins were found Trypanosoma flagella, but this is not sufficient. Immunostaining and immuno-TEM with anti-CFAP43 and 44 antibodies are needed.

Additional comments (minor)

- 1) In the human and mouse, CFAP43 and CFAP44 are expressed in the lung albeit at a low level. What happened to airway cilia of patients or mutant mouse?
- 2) Did the authors find mutations in other genes in the remaining 56 patients?

Reviewer #1 (Remarks to the Author):

Re: “Whole exome cohort study and analysis of mouse and Trypanosoma models demonstrate the importance of WDR proteins in flagellagenesis and male fertility” by Coutton et al.

The authors recruited patients showing multiple morphological abnormalities of the flagella (MMAF), which is a severe form of asthenozoospermia. 78 patients were subjected to exome analyses and 16 patients showed mutations in CFAP43 or CFAP44. Most of them are homozygous mutations and several compound heterozygous mutations. Even though each mutation frequency is extremely low, homozygous state of the patients was observed indicating the consanguineous marriage of the parents. The loss of CFAP43 and CFAP44 in human was examined by model system using mouse and Trypanosoma, and very similar phenotypes (indeed it is very surprisingly) were observed.

The authors performed functional analyses of the two genes at intensive and precise manners and the results were well described. Overall I do not have any major comment. I just point out several minor issues.

1) In the title, gene name should be shown instead of WDR proteins

This was modified.

2) Gene symbol was inconsistently used, for example; DNAH1 (in italic) gene, CFSP43 (non-italic).

We checked throughout the manuscript

3) Page 5 lines 140-143; The authors mentioned about variants as a result of negative selection. Negative or positive selection is discussed by gene-based variants (multiple variants in a gene). Indeed heterozygous state seems to be almost normal therefore it seems to be overstatement.

This was corrected.

4) Page 8 line 218; please add Fig. 4a1-3.

Done

5) Page 18, line 471; “determine” should be “determined”

This sentence was removed

6) Page 19 lines 482-484: duplicated

The duplicated sentence was removed

Reviewer #2 (Remarks to the Author):

Review

This interesting manuscript reports the discovery of two novel genes whose mutations are responsible for a male fertility disease called asthenozoospermia. Based on the analysis of patient spermatozoa, of mouse strains and of the model protist Trypanosoma brucei, the authors show that both genes are required for correct architecture of the flagellar apparatus. The paper relies on solid genetic and immunofluorescence analysis, as well as on nice electron microscopy data. The results obtained from mouse and trypanosome studies are quite spectacular. In summary, this work is novel and will be of interest for a large audience (geneticists,

cell biologists, people interested in cilia and flagella etc). However, I believe that the paper can be much improved by providing more quantitative data and by a deeper analysis of the results (see below).

Major points

1. It is intriguing that phenotypes have only been observed in spermatozoa and not in motile cilia of respiratory epithelium. I agree that the expression level is much lower, but the conserved function in trypanosomes would suggest an involvement in all kinds of motile cilia. This point should be discussed.

During their medical consultation for infertility, all subjects answered a health questionnaire focused on PCD manifestations, and none indicated suffering from any of the symptoms encountered in PCD such as cough, rhinitis, sinusitis, rhinorrhea or chronic bronchitis. This is consistent with the fact that flagellar and cilia defects are not always associated (see the review of Kavita Praveen, Erica E. Davis, and Nicholas Katsanis: Unique among ciliopathies: primary ciliary dyskinesia, a motile cilia disorder; F1000prime report, 2015, ref in the text #61). In particular, we already have shown that it is also true DNAH1 mutated patients and this was confirmed by other laboratories from studies on Chinese and Iranian cohorts (PMID: 27573432; 27798045; 28577616, ref in the text #15-17, 62).

We however found this question interesting and we performed further experiment to assess the morphology of airways cilia and the presence of clinical features of PCD in mice allowing an indirect assessment of their functional activity. Cilia morphology was evaluated with scanning electronic microscopy and clinical features by histological sections of the nasal cavity. We didn't evidence any obvious morphological difference between WT and KO mice; furthermore we did not detect any accumulation of mucus, a landmark of rhinitis and rhinorrhea. We provide two extra figures showing these data for reviewers only. These negative data are not included in the manuscript to maintain the focus on sperm and infertility.

Our observations suggest that mutations in MMAF genes are only responsible for primary infertility without other PCD features and reinforce the paradigm indicating that the sperm flagellum is assembled and organized in a specific fashion, different from the other cilia. These differences are notably illustrated by a different beating pattern and the presence of specific peri-axonemal structures. Moreover, sperm flagella are subjected to high physical stress, especially during the intra-epididymal transit that requires a considerable mechanical resistance and an adapted axonemal organization. Further work should be now performed to elucidate the differential composition and organization of these two organelles.

This point is now discussed in the discussion part.

2. A key question is unsolved in the manuscript: are CFAP43 and CFAP44 required for construction or for maintenance? The trypanosome system is famous for comparing mature and growing flagella. This analysis should be done. It is probably too challenging to do in mice.

One way answer to this question is to identify the localization of the studied proteins. For this purpose, as suggested by reviewer 2, we performed additional experiments

using the trypanosome model. In the hope of obtaining a stronger signal, we used a new 10TY Tag and produced two different endogenous tagged proteins and studied their localization by super-resolution microscopy (STED) and TEM (new figure 5 and supplementary figure 4). We showed that both proteins have a restricted localization within the flagellum, closely associated with the part of the axoneme facing the paraflagellar rod (PFR) (Fig. 5b). This localization corresponds to the doublet microtubules (DMT) 5 and 6. This location rules out a role of both proteins in intra flagellar transport (IFT) since IFT in *Trypanosoma* flagella was never described between PFR and axoneme but is circumscribed to two sets of DMTs (3 to 4 and 7 to 8) located on each side of the PFR. Moreover, IFT defects in *Trypanosoma* have been shown to induce accumulation of cytoplasmic material at the basis of the flagellum and in the flagellar pocket and a stunted flagellar growth. This was not observed upon RNAi inactivation of either genes (Fig. 6), therefore confirming that TbCFAP43 and TbCFAP44 are not involved in flagellar growth but are very likely structural proteins. This point is now clearly indicated in the discussion page 18

3. The discussion is too long (7 pages) and the reader gets lost/confused in multiple arguments and counter-arguments. It needs serious focusing. For example, a contribution of CFAP43 and CFAP44 to the architecture of the central pair is much debated without doing the right experiment that consists in showing association to the central pair by immunogold or biochemical fractionation. I agree it is a difficult experiment, especially without a good antibody, so I can understand if the authors decide not to do it. However, the discussion needs to be refocused to take this into account.

We agree with the reviewer that without the localization of the proteins the discussion was too long and speculative. Additional key experiments have now been performed (described above) which allowed us to clarify our message. The discussion was changed accordingly and the length of the discussion was significantly reduced.

4. TEM images are lovely but numbers are often vague or missing. Table S1 gives % but does not mention how many sections were looked at. I could not find any indications on the number of sections used or the proportion of the various phenotypes for Figures 5, 7 and 8. These numbers must be reported to be able to appreciate the significance of these nice images. I could not find the thickness of the sections either (important parameter knowing the repetitive nature of axoneme associated structures).

For trypanosome

section thickness : 60-80 nm

Number of sections: n=139 for TbCFAP44 RNAi, n=150 for TbCFAP43 RNAi, n=172 for WT. This information is now provided p 13, last paragraph and correspond to data presented in Figure 6.

For mouse,

section thickness: 60 nm

For figure 7 and 8 (now fig 8 and 9), this information are provided in the legend of the figures.

For Human,

section thickness : 90 nm

Number of section analyzed: 23 sections for fertile control individual, 21 sections for CFAP44 mutated patient (1 patient) and 22 sections for CFAP43 patient (1 patient). This information is now provided in the legend of table S1

5. The trypanosome data are nice but appear underused to me (also see above). Is there a motility phenotype? The authors should show a video as presented for sperm.

Video of WT trypanosome and mutants are now provided in supplementary video 3-5.

For the IFA, phase or DIC pictures should be shown to help the reader. These images are now shown in figure 5 and new figure 6 and S6

Staining patterns for TbCFAP43 and 44 look spotty. Is it related to intraflagellar transport?

As discussed above we have shown that both proteins appear on the axoneme between DMTs 5 and 6 and the PFR (Fig. 5b). Interestingly, DMTs 5 and 6 have been reported to have specific features characterized by the presence of structures likely consolidating the interaction between the axoneme and periaxonemal components and the 5-6 DMT interactions known as the 5-6 bridge. Both structures, the 5-6 bridge and the connecting proteins linking the axoneme and the PFR are known to present at regular intervals, a subcellular distribution in agreement with what is observed on the new images obtained by super-resolution microscopy and TEM. The minimum interval between two immunogold was around 25 nm for both proteins TbCFAP43 and TbCFAP44, a distance similar to the interval between two outer SUB5-6 complexes observed in the flagellum of sea urchin sperm.

Does the staining extend from base to tip in mature and growing flagella (not always clear on my images at least)?

Yes it does. This is now indicated in result section page 11.

The following sentence was added: Both proteins extended throughout the whole flagella length, as demonstrated by the myc labelling preceding the PFR at the posterior end of the flagellum (next to the kinetoplast) and going all the way to the anterior end.

How related are CFAP43 and 44?

We performed a thorough analysis of the structure of the human and Trypanosoma proteins, presented on page 10-11 in the Trypanosoma paragraph and we added a supplementary figure (S3) and a supplementary table (S3 also) to present these data.

Interestingly, CFAP43 and 44 both in human and Trypanosoma present a unique organization, with a common motif with β -strand domains in the first half (N-ter) of the four proteins and α -helical domains in the second half (C-Ter). We note that in CFAP43 and TbCFAP43, all residues between amino acids 719 and 1657 and 631 and 1446 are predicted to form α -helices, respectively. Although less spectacular, a similar organization is observed for CFAP44 and TbCFAP44. Interestingly, when the α -helices sequence from CFAP43 (amino acids 719 to 1657) is presented to

“Superfamily”, the program returns “no result” indicating that such long α -helices stretch has not yet been reported in any known fold. Altogether, these data indicate that the four tested proteins share a common structural organization suggesting that they might have a similar function.

The fate of one in RNAi conditions for the other would be interesting to know (i.e. TbCFAP44 in TbCFAP43RNAi and conversely).

These experiments were performed (see supplementary figure 7). In brief, knocking-down one protein had no impact on the expression and localization of the other one.

6. There are quite a lot of small mistakes in nomenclature or presentation of the data. It is a bit irritating. I made a list of these below but I might have missed some and I therefore strongly advise the authors to check their manuscript carefully.

The manuscript was largely modified and then corrected several times. We hope all mistakes have now been corrected

Other points

Abstract. Nowadays many authors consider cilia and flagella to be different names for equivalent structures. The authors should clearly specify if they talk about the spermatozoa flagellum or other type of cilia and flagella. The way the abstract is written shows some confusion between ciliopathies at large and PCD.

The abstract was rewritten accordingly

90. Mention the “various geographical origins” of the patients.

This information is given in the subject and controls section of Materials and methods. A short sentence has been added in the result section: “A majority of patient originated from North Africa, 46 were recruited in Tunisia, 10 in Iran and 22 in France.”

92. “indicate”

Done

96. “this cohort” instead of “the cases”

Done

124. Nomenclature: Usually genes are written in italics and proteins in standard. Please check carefully throughout the manuscript.

we did our best to correct all the mistakes

174. Which prediction softwares were used?

now indicated

187. It would be more correct to talk about the “skeleton” of cilia and flagella (no real “cyto” here)

Done

189. “complex”

Done

193. “a helical”

done

236. “reverse” genetics

done

238. Give a reference for the trypanosome model.

We already cited a reference presenting the trypanosome model « Vincensini, et al “1001 Model Organisms to Study Cilia and Flagella,” *Biology of the Cell* 103, no. 3 (2011): 109–30, doi:10.1042/BC20100104.). This is the reference **23**.

244. What is CMF7?

CFAP44 was previously reported as *T. brucei* Components of Motile Flagella 7 in reference 28 (TbCMF7), this is mentioned page 10 line 19.

247. “trypanosome” is a common name (no capital T) but “Trypanosoma” is a Latin word and should be italicised. Please check carefully throughout the manuscript.

We did our best to correct all the mistakes

251. There are two stages of trypanosomes for these experiments. Which one was used here?

We had indicated the use of bloodstream form in MM. This is now also reported in the main text.

276. The “term knock-out” does not seem appropriate as the gene is still there. Maybe “mutant” would be better.

Indels generated by CRISPR/Cas9 to made “KO animals” in now well accepted and indicate clearly that no functional protein is present in this mouse lineage.

283. On which criteria these two mouse strains were selected instead of the others? Was it a random selection (if strains really identical) or were there some practical reasons or else? In any case, it should be explained.

We have now clearly indicated that no difference were observed between different lineage after phenotyping (spermatocytogram and fertility) and lineage were chosen randomly

290. n should be provided (same lines 296 and 339)

done

for 339, n is now indicated in the legend of new figure 9

361. zebrafish and *Xenopus* are two other classic models for the study of motile cilia.

We have added this two models (page 17)

398. CFAP43 and “CFAP44”

done

480-484. Delete the duplicated sentence.

done

524. “primary”

done

848-849. Not clear what is meant by “the green staining of the entire flagellum showing that TbCFAP44 localisation is restricted to the flagellum”? The M&M mentions the use of cytoskeletons (672) so I am not sure if this statement is valid. If there were a soluble pool of TbCFAP43 or 44, it would likely be lost upon detergent extraction.

We agree with the reviewer’s remark. The Myc signal was weak and specific to cytoskeletal extracted cell only and it was not possible to evidence the presence of soluble pool with myc tagged proteins.

To test the hypothesis of the presence of soluble pool of proteins, we performed experiments with proteins tagged with 10TY1, a tag giving stronger fluorescent signal, well above background signal. Stimulated emission depletion (STED, or super-resolution) microscopy experiments were then performed on both cytoskeletal extracted cells and intact cells (whole cells). There were no differences between the two conditions, ruling out the presence of a soluble pool of TbCFAP43 or 44. This is described p.12 first paragraph.

877. remove “staining”

done

Table 1. Patients mutated for are indicated with a “+”, this is very confusing! Stars mention significance but it is not clear which numbers were compared (no data provided for healthy patients).

The table was modified.

Actually, we had compared statistical differences between MAAF due to CFAP43, CFAP44 and DNAH1 mutations versus MAAF due to uncharacterized genetic cause.

Stars indicate a significant difference $P < 0.05$

Reviewer #3 (Remarks to the Author):

In this paper, the authors have examined 78 patients showing male infertility with morphological abnormalities of sperm flagella (MMAF), by whole-exome sequencing. They identified mutations in DNAH1 (as previously reported), but in addition, they newly identified mutations in CFAP43 (10 patients) and in CFAP44 (6 patients). Analysis of mutant sperm by TEM confirmed MMAF phenotype: structural anomalies such as the absence or mis-orientation of the central pair of microtubules, abnormal organization of peripheral microtubules, disorganized mitochondrial sheath and fibrous sheath etc. They also studied CFAP43 and CFAP44 mutants in Trypanosoma and in the mouse, and found that these mutants show similar defects. In all, this is a

very solid paper reporting two evolutionarily conserved genes essential for flagella formation. However, this paper is more suited to a medical genetics journal such as *Am. J. Hum. Genet.*, because no information is available for the function of CFAP43 and CFPA44.

Additional comments (major)

1) It was speculated but CFPA43 and CFAP44 may link central pair microtubules with radial spokes, but precise localization of these proteins in the human or mouse sperm remain unknown. Are they axonemal proteins in the human or mouse? If so, where are they localized in the axoneme? Myc-tagged CFAP43 and CFAP44 proteins were found *Trypanosoma* flagella, but this is not sufficient.

We agree with the reviewer that our localization experiments were not sufficient to decipher the role of these proteins. This is particularly true since the destabilization of the axoneme can be triggered by the absence of proteins located either in the periaxonemal or axonemal structures, making it difficult to assess their function from the observed defects. Unfortunately, we could not obtain any functional antibodies. We raised 2 rabbit antisera against N-ter and C-ter epitopes (choosing sequences well conserved between mouse and human) for each of the proteins. Even after several purification steps, we could not get any band of the expected MW by Western blot nor could we get a specific signal in IF experiment in mouse or human. We feel that the only way to be sure to obtain the localisation of our proteins would now be to create some KI animals with a Cter Tag (perhaps an HA tag). Using CRISPR/Cas9 the creation of such mice is now feasible but it would take approximately one year to obtain both lineages. We felt that we could not afford that delay nor the expense and to compensate this shortcoming we decided to use another model.

The structure and organization of motile cilia or flagellum was mainly deciphered from models including the green alga *Chlamydomonas reinhardtii*, *Trypanosoma* and *Tetrahymena*, and sea urchins, the zebrafish, *Xenopus* and mouse. Such a wide selection of distant models is possible because motile cilia or flagella are built on a canonical 9 + 2 axoneme, which forms a highly organized protein network remarkably conserved during evolution. In our study, we chose to use *Trypanosoma* mutants to confirm the protein localization in the axoneme. This model has two advantages, first with a full RS3 and a fixed orientation of the central pair during flagellum beating its axoneme is closer to the mammalian flagellum than that of other models such as *Chlamydomonas*. Second, it allows fast and efficient forward and reverse genetics for an easier characterization of gene function. BlastP enabled us to identify two orthologs of CFAP44 and CFAP43 in *T. brucei* and their sequence similarities were calculated with clustal omega. Interestingly, in-depth structural analyses showed that these proteins share some striking and unique structural similarities, in particular a common general motif with β -strand domains in the N-terminal part of the protein followed by α -helical domains in the C-Ter. These results therefore confirm the homology between the human and the *T. brucei* orthologs and suggest that CFAP43 and CFAP44 could have a similar function since proteins with similar structures often have analogous functions.

To answer the reviewer queries and to precise the localization of both proteins, we

performed additional experiments; we constructed two different endogenous tagged proteins in *T. brucei*, and studied their localization by super-resolution microscopy (STED) and TEM. We show that both proteins have a restricted location within the flagellum, closely associated with the part of the axoneme facing the PFR (Fig. 5b) and corresponding to DMTs 5 and 6. Interestingly, it has been reported that around the DMTs 5 and 6, several specific structures are present, consolidating either the interaction between the axoneme and periaxonemal components or specific 5-6 DMT interactions known as the 5-6 bridge.

Overall, our results demonstrate that *TbCFAP43* or *44* and very likely their mammalian orthologs are two proteins specifically jointed to DMTs 5-6 serving to reinforce the axonemal structure. These proteins are the first to be identified at this precise localisation and these results should permit a better characterization of the protein complexes located in this area.

Immunostaining and immuno-TEM with anti-CFAP43 and 44 antibodies are needed.

As explained before we were unable to obtain any specific antibodies against the mammalian orthologs, despite numerous trials.

We do not really understand why we could not obtain any functional antibodies but we have two plausible and complementary explanations:

- 1) We noticed in that the signal obtain with the Myc tag in *T. brucei* was unusually weak. Conclusive results could only be obtained when using the 10TY1 Tag which gives a strong signal and is compatible with STED microscopy. This probably indicates that both proteins are probably scarce and this has probably contributed to the difficulties experienced.
- 2) We also have shown that the N-Ter domain of all proteins belongs to the WD40 repeat-like superfamily. The assembly of 6 to 7 WD-repeats is described to form a propeller-like structure serving as a platform for protein-protein interactions and macromolecular assembly. This suggests that *TbCFAP43* and *TbCFAP44* could be included in large protein complexes thus limiting the accessibility of the tested epitopes.

Additional comments (minor)

- 1) In the human and mouse, CFAP43 and CFAP44 are expressed in the lung albeit at a low level. What happened to airway cilia of patients or mutant mouse?

Please, refer to our response to a similar remark (#1) of reviewer 2

- 2) Did the authors find mutations in other genes in the remaining 56 patients?

The following sentence was added in the discussion p. 22:

Subsequent analysis of exome data from the remaining 56 subjects permitted to identify 7 additional candidate genes with bi-allelic mutations found in at least two individuals in a total of 23 subjects. The identified mutations and these additional genes are currently being investigated. If all these variants are confirmed to be

deleterious a diagnosis will be obtained for 45/78 patients (58%) confirming the interest of WES as a diagnostic tool for MMAF syndrome.

[redacted]

Reviewers' comments:

Reviewer #2 (Remarks to the Author):

This revised version by Coutton et al is much stronger than the original version. The authors have taken into account all my comments and addressed them very convincingly. They provide new data on the localisation of CFAP43 and CFAP44 in trypanosomes using STED and immuno-electron microscopy. They also show complete and exhaustive quantification of their electron microscopy results as well as convincing videos showing the motility phenotype. Moreover, they show clearly that CFAP43 and CFAP44 are independent from each other. The discussion has been significantly improved and the message is now clear. Overall, this really gives a new impulsion to the manuscript and makes it a significant advance compared to the recent Tang et al paper in AJHG.

The new results are really exciting because they reveal that CFAP43 and CFAP44 are associated to a subset of microtubule doublets that face an extra-axonemal structure in trypanosomes. Intriguingly, the sperm flagellum is also characterised by the presence of extra-axonemal structures that have not been observed in motile cilia in the respiratory tissues or in the brain or in the oviduct. This might provide a nice explanation to the fact that absence of CFAP43/44 has no visible impact on these populations of motile cilia. I appreciate that extra-axonemal structures are different between trypanosomes and sperm but this point is sufficiently exciting to be mentioned in the discussion. Possibly these proteins contribute to interactions between the axoneme and various extra-axonemal structures and their absence would impact on axoneme structure/construction.

I therefore strongly support acceptance of the manuscript once the minor points below are corrected.

1) The way sections are ordered is confusing to me:

-abstract: human then mouse then trypanosomes

-results: human-trypanosome-mouse

-discussion: mouse-trypanosome-human

I can see good reasons to select any of these three possibilities but please be consistent and select just one!

2) As said above, discuss the potential significance of the localisation data on the interactions between axoneme and extra-axonemal structures.

3) I would strongly advise the authors to replace the current panels of Figure 5a and Figure 6a (localisation of TbCFAP43 and TbCFAP44 using a c-myc tag) with the new figure currently presented as Figure S7. Indeed all the new localisation data is based on the 10xTy1 tag, it would make more sense to show these images in the main manuscript. In addition, the control pictures are much nicer. The current figure 5a1-4 shows a control trypanosome with a detached flagellum, this is not the normal situation! Moreover, exactly the same figure is repeated in Figure 6a (probably a mistake in assembling panels, please check carefully). This should be deleted or replaced.

4) For people not familiar with trypanosomes, it is really hard to understand which microtubule doublets are discussed (3-4-5-6-7-8 are mentioned at various places). I have two suggestions: first show a control cross-section of the flagellum where the paraflagellar rod (PFR) is visible (Fig. 6d). The current version shows a section at the very base of the axoneme where the PFR is not visible. This is confusing. Second, either annotate such a section with microtubule doublet numbers or provide a supplementary cartoon showing the way doublets are numbered in this organism.

5) PFR. This intriguing structure should be briefly described for non-trypanosome readers (what it is, its composition and function). I would be more careful when saying that there is no obvious default in PFR size (line 350). The EM pictures show a lot of electron dense material spreading from the PFR (Fig. 6d). I agree the identity is unknown at this stage but it could come from there.

6) "cytoskeleton-extracted cells" means nothing. The appropriate term is "detergent-extracted cytoskeleton" (see multiple papers from the Gull lab). Please correct throughout the manuscript.

7) Line 314. This experiment does not prove that there is no soluble pool. A cell fractionation in detergent and a western blot of all fractions would be required for that. Since it is not the main point of the paper, delete this sentence.

Reviewer #3 (Remarks to the Author):

1) Although the precise function of CFAP43 and CFAP44 still remains open, the authors have provided new data on the sub-cellular localization of these proteins. CFAP43 and CFAP44 proteins with a new tag (10TY) were found in the flagella. Interestingly, they showed restricted localization within the flagella, preferentially localized to the doublet microtubules 5 and 6. This could be a very important finding, but I have two concerns.

i) Please clarify what 10TY tag is (need to mention briefly in the text).

ii) If this tag is a long polypeptide, it would be necessary to examine if the fusion protein is functional (whether the fusion protein can rescue the corresponding mutant). I believe that this is feasible with the Trypanosome system.

2) One critical issue is the relationship between this paper and the recent paper on CFAP43 and CFAP44 by others (Tang et al., American J. Human Genetics, June 1, 2017). Both papers are quite similar, reporting i) human patients for CFAP43 and CFAP44, ii) sperm defects in human patients and knock-out mouse, iii) structural defects of mutant sperm (the lack of central pair etc). Coutton paper, however, does contain additional data: i) Trypanosoma data, ii) sub-cellular localization of CFAP43 and CFAP44 proteins, iii) expression data. In all, the Coutton paper is more comprehensive and has depth. In my opinion, the presence of the Tang et al paper alone does not preclude publication of the Coutton paper in Nat. Communic.

Reply to the reviewers.

Reviewer #2:

1) The way sections are ordered is confusing to me: -abstract: human then mouse then trypanosomes -results: human-trypanosome-mouse -discussion: mouse-trypanosome-human I can see good reasons to select any of these three possibilities but please be consistent and select just one!

This is a good point, the order human, mouse, trypanosome was maintained throughout the manuscript.

2) As said above, discuss the potential significance of the localisation data on the interactions between axoneme and extra-axonemal structures.

Following the restructuration of the manuscript we have now added a small conclusive paragraph to the discussion (p23). This point was added to this section.

3) I would strongly advise the authors to replace the current panels of Figure 5a and Figure 6a (localisation of TbCFAP43 and TbCFAP44 using a c-myc tag) with the new figure currently presented as Figure S7. Indeed all the new localisation data is based on the 10xTy1 tag, it would make more sense to show these images in the main manuscript. In addition, the control pictures are much nicer. The current figure 5a1-4 shows a control trypanosome with a detached flagellum, this is not the normal situation! Moreover, exactly the same figure is repeated in Figure 6a (probably a mistake in assembling panels, please check carefully). This should be deleted or replaced.

The figures were modified to also take into consideration reviewer 3 comments. We provide a new figure 7 which parallels the localization provided by both tags (myc and 10TY1) using two different techniques (STED and ME). The controls were modified as suggested.

4) For people not familiar with trypanosomes, it is really hard to understand which microtubule doublets are discussed (3-4-5-6-7-8 are mentioned at various places). I have two suggestions: first show a control cross-section of the flagellum where the paraflagellar rod (PFR) is visible (Fig. 6d). The current version shows a section at the very base of the axoneme where the PFR is not visible. This is confusing. Second, either annotate such a section with microtubule doublet numbers or provide a supplementary cartoon showing the way doublets

are numbered in this organism.

This is true. We therefore supplied a new figure (supplementary figure 5) illustrating the numbering of the axonemal doublets for both mouse and trypanosoma.

5) PFR. This intriguing structure should be briefly described for non-trypanosome readers (what it is, its composition and function). I would be more careful when saying that there is no obvious default in PFR size (line 350). The EM pictures show a lot of electron dense material spreading from the PFR (Fig. 6d). I agree the identity is unknown at this stage but it could come from there.

To clarify this we added in p. 13 a short referenced paragraph describing the PFR. The sentence 1.350 was modified to “There possibly were some minor modifications in PFR, however, the disruption of the axoneme was so extreme after TbCFA44 or TbCFA43 RNAi knockdown that it was difficult to assess them (data not shown).”

6) “cytoskeleton-extracted cells” means nothing. The appropriate term is “detergent-extracted cytoskeleton” (see multiple papers from the Gull lab). Please correct throughout the manuscript. This was corrected throughout the manuscript.

7) Line 314. This experiment does not prove that there is no soluble pool. A cell fractionation in detergent and a western blot of all fractions would be required for that. Since it is not the main point of the paper, delete this sentence.

This sentence was deleted.

Reviewer #3:

i) Please clarify what 10TY tag is (need to mention briefly in the text).

Information and references regarding the 10TY1 tag are now provided in the manuscript (p. 15) as below.

TY1 tag (EVHTNQDPLDGS) is a sequence of the major structural protein of the *Saccharomyces cerevisiae* Ty1 virus-like particle and is used as an epitope for immunolabelling⁴⁵. This tag has been used for numerous immunolocalizations in trypanosomes, either as 1xTY1, 3xTY1, or more recently 10xTY1⁴⁶. The 10xTY1 is 118 amino acids long, and 18

kDa. Both proteins, whatever the used tag, were found in the axoneme as substantiated by co-labelling with an antibody against the PFR structure (Supplementary Fig. 10a and 10b). This result suggested that the 10TY1 tag did not interfere with the trafficking and localization of both proteins, which extended throughout the whole flagellum length, as demonstrated by the 3myc and 10TY1 labelling preceding the PFR at the posterior end of the flagellum (next to the kinetoplast) and along the flagellum up to its anterior end.

ii) If this tag is a long polypeptide, it would be necessary to examine if the fusion protein is functional (whether the fusion protein can rescue the corresponding mutant). I believe that this is feasible with the Trypanosome system.

The legitimate concern here is that the 10TY1 tag could interfere with protein function and thus localization. We already had indicated in the manuscript that this localization was confirmed by the ME experiment realized with a myc tag, thus indicating that the described localization was supported by two distinct tags with two different techniques. We also have some additional arguments which were inserted in the new manuscript:

- 1) The TY1 tag has been used for numerous immunolocalizations in trypanosome, either as 1xTY1, 3xTY1, or more recently 10xTY1². This recent work presents concordant results indicating 10xTY1 did not alter the function and localization of the tested protein
- 2) The 10xTY1 is 118 amino acids long, and 18 kDa. This is to compare to GFP that is 239 amino acids long, and 26.82 kDa. Large tags such as GFP have been used and demonstrated to correctly localize in different axonemal components³.
- 3) More importantly and more convincingly, we already took into account a possible impact of the 10TY1 tag on protein localization by confirming the positioning of the studied proteins by electron microscopy using a different Tag (Sup. Fig S4 panel b and e). We used the myc Tag, which is only 10 amino acids long and 1.2k Da, it is therefore very unlikely to alter protein localization and if it did, it would be extraordinary that it would do so by sending the protein to the exact same localization than what is done by the 10xTY1 tag. This point was probably not clear enough and missed by reviewer 3. To clarify that point and to make it clearer to the readers we have modified our figure 8 (previously 5) and combined the STED results obtained

with the 10TY1 tag and electron microscopy results with the myc Tag confirming the localization of both proteins.

- 4) Finally we are currently using the STED microscopy for another manuscript to confirm the localization of another protein described to be axonemal. This new protein of interest was also marked with the 10TY1 tag and we clearly observe that the protein colocalizes with the axoneme (figure 1 below) thus indicating that the 10TY1 tag does not interfere with axonemal localization. This results thus confirm again the specific localization described for Cfp43 and 44 is correct.

References :

¹Albisetti et al., (2017) "Interaction between the Flagellar Pocket Collar and the Hook Complex via a Novel Microtubule-Binding Protein in Trypanosoma Brucei," *PLOS Pathogens* 13, no. 11 (November 1, 2017): e1006710

²Albisetti et al., (2017) "Interaction between the Flagellar Pocket Collar and the Hook Complex via a Novel Microtubule-Binding Protein in Trypanosoma Brucei," *PLOS Pathogens* 13, no. 11 (November 1, 2017): e1006710, <https://doi.org/10.1371/journal.ppat.1006710>.

³Haru-aki Yanagisawa et al., "FAP20 Is an Inner Junction Protein of Doublet Microtubules Essential for Both the Planar Asymmetrical Waveform and Stability of Flagella in Chlamydomonas," *Molecular Biology of the Cell* 25, no. 9 (May 1, 2014): 1472–83, <https://doi.org/10.1091/mbc.E13-08-0464>.

[redacted]

REVIEWERS' COMMENTS:

Reviewer #3 (Remarks to the Author):

As a geneticist, I am not fully convinced of authors' explanation: addition of a tag of 118 amino acid residues could have (often it does) profound effects on the function of a protein, and cell biologists tend to under-estimate such a concern. However, I do not insist this any further, because this would not be constructive under the situation. The paper can be accepted if the other reviewer is satisfied.